

# A new balance formula to estimate new particle formation rate: reevaluating the effect of coagulation scavenging

Runlong Cai and Jingkun Jiang*

State Key Joint Laboratory of Environment Simulation and Pollution Control, School of Environment, Tsinghua University, Beijing, 100084, China

*Correspondence to*: J. Jiang (jiangjk@tsinghua.edu.cn)

**Abstract.** A new balance formula to estimate new particle formation rate is proposed. It was derived from aerosol general dynamic equation in the discrete form and then converted into an approximately continuous form for analysing data from new particle formation (NPF) field campaigns. The new formula corrects the underestimation of the coagulation scavenging effect occurred in previously used formulae. It also clarifies the criterions in determining upper size bound in measured aerosol size distributions for estimating new particle formation rate. A NPF field campaign was carried out from March 7th to Apr. 7th, 2016, in urban Beijing, and a diethylene glycol scanning mobility particle spectrometer equipped with a miniature cylindrical differential mobility analyser was used to measure aerosol size distributions down to ~1 nm. 11 typical NPF events were observed during this period. Measured aerosol size distributions from 1 nm to 10 μm was used to test the new formula and those widely used ones. Previously used formulae that perform well in relatively clean atmosphere where nucleation intensity is not strong were found to underestimate the comparatively high new particle formation rate in urban Beijing because of their underestimation or neglect of the coagulation scavenging effect. Coagulation sink term is the governing component of the estimated formation rate in the observed NPF events in Beijing, and coagulation among newly formed particles contributes a large fraction to the coagulation sink term. Previously reported formation rates in Beijing and in other locations with intense NPF events might be underestimated because the coagulation scavenging effect was not fully considered, e.g., formation rates of 1.5 nm particles in Beijing are underestimated by 58.9% on average if neglecting coagulation among particles in nucleation mode.

## 1 Introduction

New particle formation (NPF) is a frequently occurring phenomenon in atmospheric environment. In a typical NPF event, gaseous precursors burst out into particles due to nucleation and lead to a rapid increase in atmospheric aerosol population. Nucleated particles can grow quickly to increase the number concentration of cloud condensation nuclei (Kerminen et al., 2012; Kuang et al., 2009; Leng et al., 2014) and thus has indirect impacts on radiative forcing and global climate (Lohmann & Feichter, 2005). Continuous growth of nucleated particles also provides increasing aerosol surface area for heterogeneous physicochemical processes, which may contribute to haze formation (Guo et al., 2014). NPF studies can trace back to the early





20th century (Aitken, 1911) and NPF events have been observed in various atmospheric environment, e.g., from city to
countryside, from desert (Misaki, 1964) to rain forest (Zhou, 2002), from continent to the ocean (Covert et al., 1992), from the
equator (Clarke et al., 1998) to polar area (Covert et al., 1996; Park et al., 2004), and from troposphere to stratosphere (Lee et
al., 2003).
Formation rate at which the growth flux past a certain diameter is a key parameter to quantitatively describe NPF events.
Different formulae have been used to estimate new particle formation rate from measured aerosol size distributions and they
mainly originate from two approaches. One is from the definition of nucleation rate (Heisler & Friedlander, 1977; Weber et
al., 1996) and the other is a population balance method (Kulmala et al., 2001; Kulmala et al., 2012). Consistency of these two
approaches was tested using a numerically simulated NPF event and a relative error of less than 20% was reported (Vuollekoski
et al., 2012). The simulated NPF event has a maximum formation rate of less than 1 $cm^{-3}$ $s^{-1}$. However, the reported formation
rates in the atmosphere vary in a large scale, e.g., approximately from $10^{-2}$ to $10^4$ $cm^{-3}$ $s^{-1}$ (Kulmala et al., 2004). Suffering
from the assumptions made in these two approaches, their validity in describing NPF events with high formation rate needs to
be further explored. A high fraction of newly formed particles is scavenged by coagulation before they grow into larger
particles. Both approaches potentially underestimate the contribution of coagulation scavenging when calculating formation
rate from measurement data. They may perform well in clean atmospheric environment where nucleation intensity is not strong
and aerosol concentration is relatively low, i.e., the coagulation scavenging effect is less important.
The effect of coagulation scavenging is more prominent when estimating formation rate of sub-3 nm particles because of their
high diffusivities and high concentrations during NPF events. Due to instrument limitations, aerosol size distributions of sub-
3 nm particles were not available in many previous NPF field campaigns. Recent developments in diethylene glycol (DEG)
condensation particle counters (CPC, Iida et al., 2009; Vanhanen et al., 2011) made it feasible to develop new scanning
mobility particle spectrometers (SMPS) for extending aerosol size distribution measurement from ~3 nm down to ~1 nm (Jiang,
et al., 2011a; Franchin *et al.*, 2016). These new spectrometers were deployed in atmospheric observations (Jiang, et al., 2011b)
and in chamber measurements (Franchin *et al.*, 2016) to study NPF. A miniature cylindrical differential mobility analyser
(mini- cyDMA, Cai et al., 2017) was developed to improve the performance of the DEG SMPS.
In many locations of China, high emissions lead to both high concentrations of gaseous precursors and high atmospheric
aerosol concentration. NPF was frequently observed even in megacities such as Beijing and Shanghai (Wu et al., 2007;
Kulmala et al., 2016; Wang et al., 2017). In most previous studies, the above population balance method was used to estimate
new particle formation rates in China. The reported formation rates of 3 nm particles and larger ones were typically in the
range of 1-10 $cm^{-3}$ $s^{-1}$ (Wang et al., 2013; Leng, et al., 2014; An et al., 2015; Qi et al., 2015). One study in Shanghai reported
a rate of 112.4 to 271.0 $cm^{-3}$ $s^{-1}$ for the formation of 1.5 nm particles inferred from a DEG CPC (Xiao et al., 2015). For these
intense NPF events, the above balance approach may underestimate the coagulation scavenging effect and thus lead to



underestimation in the reported formation rate. In addition, applying new SMPSs to measure aerosol size distributions down
to ~ 1 nm will help to better quantify the formation rate and its governing factors in typical locations of China.
When estimating new particle formation rates, various particle size ranges were used in previous formulae. The definition
approach tries to limit the size range towards the minimum detected diameter (Kuang *et al.*, 2008; Weber, et al., 1996), while
studies with the population balance method have used various size ranges. Some studies used the aerosol size distributions
from the minimum detected diameter up to 25 nm (Kulmala et al., 2001; Dal Maso et al., 2005; Wu et al., 2007; Wang et al.,
2013). Kulmala et al. (2004) recommended the upper size bound as the maximum size that the critical cluster can reach during
a short time interval of growth. There are also studies using narrower size range such as from 3 nm to 6 nm (Sihto et al., 2006;
Paasonen et al., 2009; Wang et al., 2011; Vuollekoski et al., 2012) and from 1.34 nm to 3 nm (Xiao et al., 2015). In principle,
the estimated formation rates may vary when different particle size ranges are used. Assumptions made while deriving these
formulae should be fully considered when proposing criterions to choose particle size range.
In this study, a new population balance formula for estimating new particle formation rate is derived from aerosol general
dynamic equation to properly account for the effect of coagulation scavenging, especially for analysing intense NPF events.
A NPF field campaign is carried out in Beijing. Aerosol size distributions down to ~ 1 nm are measured using the DEG SMPS
equipped with the mini- cyDMA. Data from this campaign and from literature are used to test the new formula and other
widely used formulae. Different formulae are compared and their applicability in analysing intense NPF events are addressed.
Criterions to choose particle size range for formation rate estimation are proposed and evaluated. Governing components of
the new formation rate in Beijing are discussed and compared to those from other locations in the world.
**2 Theory**
The new formula based on definition of droplet current and aerosol general dynamic equation (see Appendix A for its
derivation) is shown in Eq. (1),
$$J_k = \frac{\mathrm{d}N_{[d_k,d_u)}}{\mathrm{d}t} + \sum_{d_g=d_k}^{d_{u-1}} \sum_{d_i=d_{min}}^{+\infty} \beta_{(i,g)} N_{[d_i,d_{i+1})} N_{[d_g,d_{g+1})} - \frac{1}{2} \sum_{d_g=d_k}^{d_{u-1}} \sum_{\substack{d_i^3+d_{j+1}^3=d_g^3 \\ d_{i+1}^3+d_j^3=d_g^3 \\ d_i,d_j \geq d_{min}}} \beta_{(i,j)} N_{[d_i,d_{i+1})} N_{[d_j,d_{j+1})} + n_u \cdot GR_u \qquad (1)$$
where $J_k$ is the formation rate of particles at size $d_k$; $N$ is particle number concentration and $N_{[d_k,d_u)}$  is defined as total number
concentration of particles ranged from $d_k$ to $d_u$ (not included); $d_i$ refers to the lower bound of each measured size bin; $\beta_{(i,g)}$ is
the coagulation coefficient when particles with the diameter of of $d_i$ collides with particles with the diameter of $d_g$; $n$ is particle
size distribution function which equals to $\mathrm{d}N/\mathrm{d}d_p$; and $GR_u$ is particle growth rate at $d_u$, i.e., $\mathrm{d}d_u/\mathrm{d}t$. $d_u$ is the upper bound of the
size range for calculation. $d_{min}$ is the size of minimum cluster in theory and the lowest size limit of measuring instrument in





practice. The last three terms in the right hand side (RHS) of Eq. (1) are coagulation sink term (*CoagSnk*), coagulation source
term (*CoagSrc*) and condensational growth term, respectively.
The two assumptions of Eq. (1) are that (a) transport, dilution, primary emission and other losses except for coagulation loss
in the size range from $d_k$ to $d_u$ are comparatively negligible; (b) when deriving the fourth term in the RHS of Eq. (1), net
coagulation (net result of both formation and scavenging due to coagulation) of any particle larger than $d_u$ with other particles
is negligible. These two assumptions above are also the criterions to determine $d_u$. The mathematical expression of population
balance in Eq. (1) in discrete form is illustrated by Fig. 1. Time rate of change of particles at $d_k$ is equal to source minus sink.
Source are the condensational flux into $d_k$ ($J_k$) and formation due to coagulation among smaller particles/clusters (*CoagSrc$_k$*).
Sink are the condensational flux out of $d_k$ ($J_{k+1}$) and scavenging due to coagulation with other particles/clusters (*CoagSnk$_k$*).
Nucleation rate, $I$, is defined as $J_k$ when $d_k$ is the size of the critical cluster (nuclei). Equation (1) is obtained by adding these
single population balance equations up from $d_k$ to $d_u$, converting it from the discrete form into the continuous form, and
approximating $J_u$ with the product of measured $n_u$ and $GR_u$. Note that Eq. (1) is still an approximate formula of particle
formation rate because *CoagSnk* and *CoagSrc* are calculated by size bins and coagulation effect of particles smaller than $d_{min}$
is not included. For rigorous mathematical derivation and detailed illustration, please refer to Appendix A.
The population balance method proposed in previous study is shown in Eq. (2) (Kulmala et al., 2001; Kulmala et al., 2012),
$$J_k = \frac{dN_{[d_k,d_u)}}{dt} + CoagS_m \cdot N_{[d_k,d_u)} + \frac{N_{[d_k,d_u)}}{(d_u - d_k)} \cdot GR_{[d_k,d_u)} \tag{2}$$
where coagulation sink, $CoagS_m$, is defined as Eq. (3).
$$CoagS_m = \int_0^{+\infty} \beta_{(i,m)} n_i \, dd_i \tag{3}$$
The subscript m corresponds to the representing diameter, $d_m$, for particles ranged from $d_k$ to $d_u$. $d_m$ is often estimated as the
geometric mean diameter of $d_k$ and $d_u$. Equation (1) and (2) look similar because they are both derived from the general dynamic
equation, while their detailed differences are illustrated in Appendix B.
The definition approach to calculate new particle formation rate is shown in Eq. (4) (Heisler & Friedlander, 1977; Weber et
al., 1996; Iida et al., 2006; Kuang et al., 2008; Kuang et al., 2012).
$$J_k = n_k \cdot GR_k \tag{4}$$
Equation (4) focuses on the flux into $d_k$ and is theoretically correct in continuous space of particle diameter. However, when
applying Eq. (4) in practice, size distribution of particles small than $d_k$ is required, which is difficult to obtain. Usually diameter
bins larger than $d_k$ are used to estimate particle formation rate when using the practical expression of Eq. (4) (e.g., Eq. (9) as
defined in section 4.3). As illustrated in Fig. 1, such approximation essentially neglects the first three terms in the RHS of Eq.
(1), and may lead to underestimation of particle formation rate because of neglecting the coagulation scavenging effect
especially when analysing intense NPF events.



## 3 Experiment

A NPF field campaign was carried out in Beijing. The observation period is from March 7[th] to April 7[th], 2016. The monitoring

site locates on the top floor of a four-storey building in the centre of the campus of Tsinghua University. Tsinghua situates in

the northwestern urban area of Beijing and the fourth-ring road is ~2 km away to the south of the monitoring site. The site has

been a $PM_{2.5}$ monitoring station since 1999 (He et al., 2001; Cao et al., 2014) and there is no tall building nearby. Potential

pollution sources around are the three cafeterias on campus that may produce cooking aerosol during meal time, locate ~170

m away on the northeast, ~170 m away on the north, and ~350 away on the northwest, respectively.

A DEG SMPS equipped with a mini- cyDMA specially designed for classification of sub-3 nm particles was deployed to

measure particles in the size range of 1-5 nm (Cai et al., 2017). A particle size distribution system, including a SMPS with a

TSI nano DMA, a SMPS with a TSI long DMA and an aerodynamic particle sizer, was used to measure particles in the size

range of 3 nm to 10 μm in parallel (Liu et al., 2016). Other instruments whose data are not used in this analysis are not listed

here.

A C++ program was used to invert particle size distribution from raw counts while incorporating diffusion losses inside the

sampling tube, charging efficiencies of the bipolar neutralizers, penetration efficiencies of DMAs, and detection efficiencies

of CPCs. Particle density was assumed to be 1.6 g/cm$^3$ according to local observation results (Hu et al., 2012). Coagulation

efficiency was assumed to be unity and temperature was assumed to be a constant value of 285 K, the average temperature

during the observation period.

## 4 Results and discussion

### 4.1 Upper size bound for formation rate calculation

New particle formation rates using different upper size bound, $d_u$ of 3 nm, 6 nm, 10 nm and 25 nm were calculated. A varying

upper size bound, $d_b$, was also tested. It is theoretically defined as the maximum size that particles formed by nucleation have

reached and is practically determined as the largest size bin in the size range from 3 nm to 25 nm whose frequency density

(particle size distribution), $dN/d\log d_p$, was larger than 28,000 #/cm$^3$. Here 28,000 was determined according to measured

particle size distribution and the value might be campaign specific or even event specific. Note that $d_u$ is equal to 25 nm rather

than $d_b$ when calculating $dN/dt$ to avoid potential influence of varying size range on particle number concentration. Fig. 2(a)

indicates that $d_b$ is almost the boundary of particles formed due to nucleation. Thus, the estimated $J_{1.5}$ using $d_b$ is regarded as

a relatively credible value when compared to others.

As shown in Fig 2(b), estimated $J_{1.5}$ using $d_u$ equal to $d_b$ and a constant value of 25 nm are almost the same (the mean relative

error is 2.2%). Maximum difference between these two choices is ~10% which appears before 8:00 when $d_b$ is less than 5 nm

and the number concentration of sub-25 nm particles is ~2 times of sub-6 nm particles and ~3 times of sub-3 nm particles. It





indicates the influence of non-freshly nucleated particles on estimating $J_{1.5}$ is not important because their comparatively low
diffusivities even though their concentration is comparatively high at the beginning of NPF events. Estimated $J_{1.5}$ using $d_u$ of
6 nm and 10 nm is in good consistency with that using $d_b$ before 10:00 (the mean relative errors are 4.8% and 2.6%,
respectively). However, when particles formed by nucleation grow beyond, calculated $J_{1.5}$ is underestimated when using 6 nm
and 10 nm as the upper bound. For example, the mean relative errors of estimated $J_{1.5}$ using $d_u$ of 6 nm and 10 nm between
10:30 and 15:00 are 18.6% and 12.8%, respectively. When calculating $J_{1.5}$ using 3 nm as $d_u$, an average 47% underestimation
was found for this event.
The reason of underestimation when using smaller $d_u$ can be illustrated by Fig. 2(c). $J_u$ is estimated by $n_u \cdot GR_u$ in Eq. (1).
This estimation may be not accurate when $d_u$ is small because the assumption that net coagulation of any particle larger than
$d_u$ with other particles is negligible may be violated. As illustrated in the derivation of Eq. (1) in Appendix A, a nearly zero $J_u$
is preferred when using Eq. (1). However, as shown in Fig 2(c), estimated $J_3$ is still a large fraction compared to $J_{1.5}$, while $J_6$
and $J_{10}$ are 27.8% and 17.6% of $J_{1.5}$ on average between 10:30 and 15:00, respectively. Although $J_u$ is approximated by
$n_u \cdot GR_u$ rather than simply neglected, this approximation may still lead to uncertainties.
Since $J_{1.5}$ estimated by the varying $d_b$ and a constant value of 25 nm is almost the same with an acceptable relative error even
under the interference of non-freshly nucleated particles, 25 nm was adopted as the upper bound for calculating $J$ in this study.
Since $d_u$ is "proper large" as defined by the two criterions, it is also reasonable to neglect $J_u$ for simplicity. It should be clarified
that 25 nm is not necessarily valid for all other studies, because the upper bound should be determined by the two criterions
and can be campaign specific. However, it can be concluded that a very small upper bound such as 3 nm is not recommended
because particles formed by nucleation surely grow larger than 3 nm in a typical NPF event while intense primary emission of
particles around 3 nm is rarely observed in the atmosphere (unless near the emission sources).
**4.2 Comparison with previous formulae**
Equation (5) is a widely used balance formula to estimate formation rate in previous studies (Kulmala et al., 2001; Dal Maso
et al., 2005; Wu et al., 2007; Shen et al., 2011; Wang et al., 2013),
$$J_{1.5} = \frac{\mathrm{d}N_{[1.5,25]}}{\mathrm{d}t} + N_{[1.5,25]} \sum_{d_i=1.5\mathrm{nm}}^{+\infty} \beta_{(i,8)} N_i + \frac{N_{[1.5,25]}}{(25-1.5)\,\mathrm{nm}} \cdot GR_{[1.5,25]} \tag{5}$$

where $N_i$ is the number concentration of size bin i. Corresponding to those in Eq. (2), $d_u$ is 25 nm and $d_m$ is 8 nm in Eq. (5). By
comparing Eq. (5) with Eq. (1), it can be concluded that Eq. (5) estimates $CoagSnk$ using a representative $CoagS_m$ and neglects
$CoagSrc$.





When calculating $CoagS_m$, particles smaller than $d_m$ (Kulmala et al., 2012) or even $d_u$ are neglected in some previous studies.
Corresponding formulae are shown in Eq. (6) and Eq. (7), respectively. The only difference among Eq. (5), Eq (6), and Eq. (7)
is the lower bound when calculating $CoagS_m$ in the second term in the RHS of these equations.
$$J_{1.5} = \frac{\mathrm{d}N_{[1.5,25)}}{\mathrm{d}t} + N_{[1.5,25)} \sum_{d_i=8\mathrm{nm}}^{+\infty} \beta_{(i,8)} N_i + \frac{N_{[1.5,25)}}{(25-1.5)\,\mathrm{nm}} \cdot GR_{[1.5,25)}$$
(6)

$$J_{1.5} = \frac{\mathrm{d}N_{[1.5,25)}}{\mathrm{d}t} + N_{[1.5,25)} \sum_{d_i=25\mathrm{nm}}^{+\infty} \beta_{(i,8)} N_i + \frac{N_{[1.5,25)}}{(25-1.5)\,\mathrm{nm}} \cdot GR_{[1.5,25)}$$
(7)

The upper bound, $d_u$, is selected as 6 nm is some recent studies (Sihto et al., 2006; Riipinen et al., 2007; Paasonen et al., 2009;
Wang et al., 2011; Vuollekoski et al., 2012; Wang et al., 2015), shown in Eq. (8).
$$J_{1.5} = \frac{\mathrm{d}N_{[1.5,6)}}{\mathrm{d}t} + N_{[1.5,6)} \sum_{d_i=1.5\mathrm{nm}}^{+\infty} \beta_{(i,3)} N_i + \frac{N_{[1.5,6)}}{(6-1.5)\,\mathrm{nm}} \cdot GR_{[1.5,6)}$$
(8)

It should be clarified that $d_k$ in Eq. (5)-(8) is usually 3 nm in previous studies due to the absence of sub-3 nm particle size
distributions, and $d_m$ in Eq. (8) is 4 nm rather than 3 nm in previous studies because 4 nm is almost the geometrical mean
diameter of 3 nm and 6 nm. Particles smaller than 6 nm were neglected in some studies, although its uncertainties will not be
discussed here. The expression of condensational growth term, i.e., the third term in the RHS of Eq. (8) varies with studies,
however, it does not influence the generality of the following discussion.
In previous studies, several size bins larger that $d_k$, typically 3 nm, were adopted when using the practical formula of the
definition approach (Weber et al., 1996; Kuang et al., 2008), while here the size range from 1.5 nm to 2.5 nm is applied to
estimate $J_{1.5}$ as shown in Eq. (9).
$$J_{1.5} = \frac{N_{[1.5,2.5)}}{(2.5-1.5)\,\mathrm{nm}} \cdot GR_{[1.5,2.5)}$$
(9)

Estimated $J_{1.5}$ values using Eq. (1) and Eq. (5)-(9) on March 13th are shown in Fig. 3. $d_k$, $d_u$, and $d_{min}$ are 1.5 nm, 25 nm, and
1.3 nm, respectively, when using Eq. (1). It can be concluded that except for Eq. (8), other formulae significantly underestimate
$J_{1.5}$ compared to Eq. (1). By comparing contribution of each terms in the RHS of Eq. (1) and Eq. (5)-(9), it was found that
underestimation of formation rates is mainly caused by the underestimation of $CoagSnk$. Equation (9) simply neglects $CoagSnk$
as well as other terms (d$N$/d$t$ and $CoagSrc$) compared to Eq. (1), so its result is the lowest among six formulae. Equation (5)
estimates $CoagSnk$ using an average $CoagS_m$, which lead to underestimation because $CoagS$ at 8 nm happens to be smaller
than those at most other diameters in the size range from 1.5 nm to 25 nm, as illustrated in Appendix B. Equation (6) and (7)
neglects particles smaller than 8 nm and 25 nm when calculating $CoagS_m$, respectively. Such simplification may be reasonable
for relative clean atmosphere where nucleation intensity is not strong, however, these approximations are not suitable for
analysing typical NPF events in Beijing where coagulation among nucleation mode particles is a major proportion of $CoagSnk$.





$J_{1.5}$ estimated using Eq. (8) seems to agree well with that estimated using Eq. (1), however, this agreement is because that the
underestimation of *CoagSnk* is smaller when using an average $CoagS_m$ in a narrower size range and this underestimation is
coincidentally cancelled out by the overestimation of formation rate caused by neglecting *CoagSrc*.
The importance of coagulation scavenging among newly formed particles due to nucleation is illustrated in Fig. 4. Scavenging
due to coagulation with particles smaller than $d_p$ is neglected, as mathematically defined in the formula in Fig. 4(a). *CoagSnk*
increases rapidly with the decrease in $d_p$ rather than maintain an approximately constant value during NPF periods, indicating
coagulation among nucleated particles contribute a considerable fraction to *CoagSnk* in Beijing. The necessity of sub-3 nm
particle size distribution is also demonstrated, which means estimated $J_3$ may also be underestimated due to the absence of
sub-3 nm data, as illustrated in Appendix B. Approximation of *CoagSnk* estimated using a representative $CoagS_m$ is also shown
in Fig. 4(b), indicating the underestimation of new particle formation rate when applying Eq. (5) to analyse NPF events in
Beijing. However, calculated *CoagSnk* on a non-NPF event day as well as at non-NPF periods on NPF day is almost unaffected
by the coagulation scavenging effect of particles in nucleation mode (smaller than 25 nm), because number concentration of
nucleation mode particles at non-NPF time is comparatively low.
**4.3 Characteristics of estimated formation rate in Beijing**
For NPF events observed in the Beijing campaign, *CoagSnk* is a governing component of the estimated $J_{1.5}$. Estimated
formation rate on March 13[th] and the four terms in the RHS of Eq. (1), i.e., d$N$/d$t$, *CoagSnk*, *CoagSrc*, and the condensational
growth term, are shown in Fig. 5. *CoagSnk* is almost the same with the estimated $J_{1.5}$ in Beijing, while the difference between
them is mainly due to d$N$/d$t$ whose absolute value is comparatively higher at the beginning and the end of the NPF event. The
condensational growth term, $n_u \cdot GR_u$, is negligible compared to other terms, which is reasonable since $J_u$ is supposed to be
unimportant when determining $d_u$ in Eq. (1). The governing role of *CoagSnk* in estimated formation rate in Beijing emphasizes
the importance of fully considering the coagulation scavenging effect among particles formed by nucleation. Equation (5)-(9)
may fit well in relatively clean atmospheric environment where new particle formation rate is comparatively low, such as in
Hyytiälä, and agreement of Eq. (8) and Eq. (9) has been reported in a numerically simulated NPF event in which $J_3$ is less than
1 cm$^{-3}$s$^{-1}$ (Vuollekoski et al., 2012). However, problems appear when applying them in urban Beijing because of
underestimating the governing fraction of estimated $J_{1.5}$, i.e., *CoagSnk*.
Coagulation sink, *CoagS*, is not the major reason for the governing role of *CoagSnk* in Beijing. It is generally considered that
the atmosphere in typical urban area in China, such as Beijing, is comparatively polluted. However, observed NPF events
mainly occurs on clean days when the air mass comes from north or northwest of Beijing. Mean $PM_{2.5}$ mass concentration
reported by the nearest national monitoring station, Wanliu station, was 10.4 μg/cm$^3$ during all NPF events in this campaign.
Aerosol surface area concentration is characterized by Fuchs surface area, $A_{Fuchs}$ (McMurry, 1983), and condensation sink, *CS*



(Kulmala et al., 2001), which are often used to examine the coagulation scavenging effect. The positive correlation between
$A_{Fuchs}$ and $CS$ is illustrated in McMurry et al. (2005), while $CS$ can be regarded as the $CoagS$ of sulphuric acid molecules. Fig.
6(a) shows the comparison of $A_{Fuchs}$ and $CS$ in Beijing to those in other locations around the world. $A_{Fuchs}$ and $CS$ during NPF
events in this study are higher than those in Hyytiälä, similar to those observed in Boulder, and lower than those in Atlanta
and Mexico City. This indicates coagulation sink in urban Beijing on NPF days is in common range rather than higher than
most other places around the world.
Nucleation intensity in urban Beijing, characterized by number concentration of particles larger than 3 nm, is found to be
higher than those in Hyytiälä and Atlanta (as shown in Fig. 6(b)). Number concentration of sub-3 nm particles is not included
to maintain comparability. Although $A_{fuchs}$ and $CoagS$ represent the relative importance of the coagulation scavenging effect
(McMurry, 1983; Kulmala et al., 2001), it is the $CoagSnk$ that reflects the number of particles lost due to coagulation
scavenging in the size range of $d_k$ to $d_u$. Equation (1) shows that $CoagSnk$ is approximately proportional to the square of particle
number concentration. This explains the governing status of $CoagSnk$ in estimated formation rates in urban Beijing with intense
NPF events.
Fig. 7 further illustrate the underestimation in new particle formation rates in China due to previously used formulae, especially
for Eq. (7) which neglects coagulation among sub-25 nm particles and Eq. (9) which simply neglects net coagulation effect.
Mean $J_{1.5}$ estimated in this study using Eq. (5), Eq. (7), and Eq. (9) are 87.1%, 41.1% and 15.7% of that estimated by Eq. (1).
Mean $J_3$ estimated using Eq. (5), Eq. (7), and Eq. (9) are 87.3%, 49.9% and 30.7% of that estimated by Eq. (1). $J_3$ reported in
previous studies in urban Beijing (Wu et al., 2007; Yue et al., 2009; Wang et al., 2013; Wang et al., 2015), Shanghai (Xiao et
al., 2015) and in Shangdianzi, the regional background station of North China Plain (Shen et al., 2011; Wang et al., 2013), are
also shown in Fig. 7. Higher formation rates are anticipated if the coagulation scavenging effect are fully considered when
analysing these NPF events. Note that sub-3 nm particles is also accounted when calculating $J_3$ in this study, while not in
previous ones except for the campaign in Shanghai.
**4 Conclusions**
A new balance formula to estimate new particle formation rate derived from aerosol general dynamic equation was proposed.
The new formula estimates the effect of coagulation scavenging better compared to previously used ones. Two criterions in
determining the upper bound for calculation were proposed. A NPF campaign in urban Beijing was carried out in spring of
2016. Aerosol size distributions down to ~1 nm was measured and used to test the new formula and those widely used ones in
previous studies. It was found that formation rates in urban Beijing are underestimated to different extent in previously used
formulae, and the underestimation of the coagulation scavenging effect (corresponding to coagulation sink term) is the major
reason. Coagulation among particles in nucleation mode was found to be important when estimating the coagulation




scavenging effect in urban Beijing and neglecting it can lead to an average 58.9% underestimation in the estimated formation
rate of 1.5 nm particles. Coagulation sink term is the governing component of the estimated formation rate in urban Beijing.
Although higher than those in relative clean atmosphere such as in Hyytiälä, coagulation sink (expressed in the form of Fuchs
surface area and condensation sink) in urban Beijing on NPF days is lower than those reported in Atlanta and Mexico City.
However, number concentration of particles formed due to nucleation in urban Beijing is comparatively high, which lead to
high coagulation loss. Formulae used in previous studies may perform well when describing relative weak NPF events in clean
atmosphere, while they underestimate the coagulation scavenging effect when analysing intense NPF events. Formation rates
reported in previous studies for urban Beijing and other locations with intense NPF events might be underestimated because
of their underestimation or neglect of the coagulation scavenging effect.
**Appendix A**
**Derivation of nucleation rate from aerosol general dynamic equation**
Nucleation rate is the rate at which particles grow past the size of the critical cluster (nuclei). However, a more specific and
microscopic definition of nucleation rate is needed for any further calculation, and it should be easily and unambiguously
transferred into a mathematical expression. Here we adopt the definition based on droplet current (Eq. 10.1, Friedlander 2000):
$$J_g = \beta_{(1,g-1)} N_1 N_{g-1} - \alpha_g s_g N_g \,. \qquad\qquad (A1)$$
Formation rate, $J_g$, is the excess rate of the passage from g-1 (cluster or particle with g-1 molecules) to g by condensation over
the passage from g to g-1 by evaporation. If g is the size of the critical cluster, $J_g$ is defined as nucleation rate, $I$. $N_g$ is the
number concentration of cluster g; $\beta_{(i,j)}$ is the coagulation coefficient of i and j, and it can be theoretically estimated by diameter
of i and j (Eq. 13.56, Seinfeld & Pandis 2006); $\alpha_g$ is the monomer evaporation flux from g; and $s_g$ is the effective surface area
of g for evaporation. Only formation due to condensational growth is considered in the definition of Eq. (A1), while formation
due to coagulation of smaller clusters is not taken into account. This is based on the assumption that critical clusters are mainly
formed by condensational growth of sulfuric acid and other chemical species. The formation of critical cluster by coagulation
does not influence the generality of the following derivation and can be readily incorporated, and it will be clarified at the end
of Appendix A.
The other basic equation for the derivation is the general dynamic equation in the discrete form (Eq. 11.3, Friedlander 2000),
$$\frac{\mathrm{d}N_g}{\mathrm{d}t} = \frac{1}{2}\sum_{\substack{i+j=g \\ i,j\geq 2}} \beta_{(i,j)} N_i N_j - \sum_{i=2}^{+\infty} \beta_{(i,g)} N_i N_g + \beta_{(1,g-1)} N_1 N_{g-1} - \beta_{(1,g)} N_1 N_g - \alpha_g s_g N_g + \alpha_{g+1} s_{g+1} N_{g+1}\,. \qquad (A2)$$
As shown in Eq. (A2), time rate of change of cluster or particle number concentration, $\mathrm{d}N_g/\mathrm{d}t$ in the left-hand side (LHS), is
determined by formation due to coagulation of smaller clusters and (or) particles, coagulation scavenging with pre-existing





clusters and particles, condensational growth from g-1 and to g+1, and evaporation to g-1 and from g+1, corresponding to the
six terms in the right-hand side (RHS) of Eq. (A2), respectively. The evaporation terms (corresponding to the fifth and sixth
terms in the RHS) may be zero or nearly zero when g is large, however, their exact values have no influence on derivation. An
important assumption to be noted is that meteorological transport, dilution, primary emission of g and other losses (e.g., wall
loss) are not included in Eq. (A2).
Notice that the last four terms in the RHS of Eq. (A2) are equal to $J_g - J_{g+1}$ by substituting Eq. (A1) in. Replacing subscript g
with the critical cluster size, k, we have:
$$I := J_k = \frac{dN_k}{dt} + \sum_{i=2}^{+\infty} \beta_{(i,k)} N_i N_k - \frac{1}{2} \sum_{\substack{i+j=k \\ i,j \geq 2}} \beta_{(i,j)} N_i N_j + J_{k+1} .$$
(A3)

The expression of Eq. (A3) is similar to Eq. (A6) in Kuang et al. (2012), which was also obtained using the balance method.
$J_{k+1}$ is usually a relatively large term in Eq. (A3), and it can be accounted for by iteration. Equation (A5) is obtained by
summing Eq. (A3) up from subscript k to u-1 as shown in Eq. (A4), where u is the particle size at the upper bound of the
concerned size range.
$$I - J_{k+1} = \frac{dN_k}{dt} + \sum_{i=2}^{+\infty} \beta_{(i,k)} N_i N_k - \frac{1}{2} \sum_{\substack{i+j=k \\ i,j \geq 2}} \beta_{(i,j)} N_i N_j$$

$$J_{k+1} - J_{k+2} = \frac{dN_{k+1}}{dt} + \sum_{i=2}^{+\infty} \beta_{(i,k+1)} N_i N_{k+1} - \frac{1}{2} \sum_{\substack{i+j=k+1 \\ i,j \geq 2}} \beta_{(i,j)} N_i N_j$$

$$...... = ......$$

$$J_{u-1} - J_u = \frac{dN_{u-1}}{dt} + \sum_{i=2}^{+\infty} \beta_{(i,u-1)} N_i N_{u-1} - \frac{1}{2} \sum_{\substack{i+j=u-1 \\ i,j \geq 2}} \beta_{(i,j)} N_i N_j$$
(A4)

$$I = \frac{d\sum_{g=k}^{u-1} N_g}{dt} + \sum_{g=k}^{u-1} \sum_{i=2}^{+\infty} \beta_{(i,g)} N_i N_g - \frac{1}{2} \sum_{g=k}^{u-1} \sum_{\substack{i+j=g \\ i,j \geq 2}} \beta_{(i,j)} N_i N_j + J_u$$
(A5)

In the RHS of Eq. (A5) are the time rate of change of the particle concentration, the coagulation sink term, the coagulation
source term and the condensational growth term, respectively. Note that when particle u is large enough, $J_u$ is nearly zero, i.e.,
$\lim_{u \to \infty} J_u = 0$ , because of their negligible condensational growth and low number concentration compared to those of freshly
nucleated small particles. Equation (A6) is obtained by replacing the upper bound, u, with infinite and further simplified by
combining the second and third term in the RHS of Eq. (A5).
$$I = \frac{d\sum_{g=k}^{+\infty} N_g}{\partial t} + \frac{1}{2} \sum_{g=k}^{+\infty} \sum_{i=k}^{+\infty} \beta_{(i,g)} N_i N_g$$
(A6)

Theoretically, Eq. (A6) can be used to estimate $I$ since each term in the RHS can be calculated. However, the validity of Eq.
(A6) faces higher risk of violation when applied in real atmosphere due to non-negligible primary emission sources, since Eq.





(A6) is a balance equation for the whole aerosol population rather than a limited size range of the nucleation mode. It's both
more cautious and efficient to use Eq. (A5) with a proper particle size u and a reasonable estimation of $J_u$.
When using measured particle size distribution to estimate $I$, Eq. (A5) has to be converted from the discrete form into the
continuous form, i.e., Eq. (A7). Since measured size bins are finite, Eq. (A7) is expressed in the summation form rather than
the integration form. Practically Eq. (A7) is only an estimation of Eq. (A5) because coagulation is calculated by size bins,
while particles sizes in each size bin are not exactly the same as the representing diameter, $d_g$.
$$I = \frac{dN_{[d_k,d_u)}}{dt} + \sum_{d_g=d_k}^{d_{u-1}} \sum_{d_i=d_{min}}^{+\infty} \beta_{(i,g)} N_{[d_i,d_{i+1})} N_{[d_g,d_{g+1})} - \frac{1}{2} \sum_{d_g=d_k}^{d_{u-1}} \sum_{\substack{d_i^3+d_{j+1}^3=d_{g}^3 \\ d_{i+1}^3+d_j^3=d_{g}^3 \\ d_i,d_j \geq d_{min}}} \beta_{(i,j)} N_{[d_i,d_{i+1})} N_{[d_j,d_{j+1})} + J_u \tag{A7}$$
$d_{min}$ is theoretically the minimum cluster size. Note that the size bin from $d_{u-1}$ to $d_u$ is denoted by subscript u-1, so the upper
bound of the size range for calculation is $d_u$. $N_{[d_k,d_u)}$ is defined as the number concentration in the size range from $d_k$ to $d_u$
(not included), corresponding to $\sum_{g=k}^{u-1} N_g$ in the discrete from. $d_u$ is a "proper large" size at which diameter $J_u$ is negligible
compared to the sum of the others three terms in the RHS of Eq. (A7). "Proper large" is defined by the following two criterions:
the one is $d_u$ shouldn't be too large so that the calculated nucleation rate is non-negligibly affected by transport and primary
emissions; the other is $d_u$ shouldn't be too small so that the calculated nucleation rate is underestimated because $J_u$ is still too
large to be neglected or to be estimated by growth rate (as illustrated in the following paragraph). These two criterions seem
to be contradictory, however, as illustrated in Fig. 4, calculated nucleation rate is usually not sensitive to the upper bound
because $J_u$ decreases rapidly with the increase of $d_u$ since the freshly nucleated particles are usually in a relatively narrow size
range, especially during strong NPF events.
The fourth term in the RHS of Eq. (A7), $J_u$, is usually so small that it can be simply neglected when $d_u$ is proper large. However,
an approximate term is recommended for better estimation. Here we introduce a sufficient but possibly unnecessary condition
that net coagulation effect between any particle larger than $d_u$ and other particles can be neglected. Define $N_{[d_u,d_u+\Delta d)}\big|_t$ as
number concentration of particles in a narrow size range from $d_u$ to $d_u+\Delta d$ at time t. After a very short time d$t$, these particles
grow into the size range from $d_u+dd$ to $d_u+\Delta d+dd$, which is based on the assumption that diameter growth is equal for different
particles in such narrow size and time range, while number concentration remains the same since there is no particle loss.
Particles in the size range from $d_u+\Delta d$ to $+\infty$ at time t grow up to the size range from $d_u+\Delta d+dd$ to $+\infty$, correspondingly. And
since the size range is narrow enough, it's reasonable to assume that concentration of particles is equally distributed in the size
range from $d_u$ to $d_u+\Delta d+dd$, i.e.,
$$\frac{N_{[d_i,d_j)}\big|_{t+dt}}{N_{[d_m,d_n)}\big|_{t+dt}} = \frac{d_j-d_i}{d_n-d_m}, \text{ for any } d_i,d_j,d_m,d_n \in [d_u,d_u+\Delta d+dd). \tag{A8}$$





Particle size distribution function, $n$, and growth rate, $GR$, are defined as Eq. (A9) and A(10), respectively. Equation (A11) is
obtained by combining Eq. (A6), Eq. (A8), Eq. (A9), and Eq. (A10).
$$n_u = \frac{dN}{dd}\bigg|_{d_u} = \lim_{\Delta d \to 0} \frac{N_{[d_u,d_u+\Delta d]}}{\Delta d} \tag{A9}$$

$$GR_u = \frac{dd}{dt}\bigg|_{d_u} \tag{A10}$$

$$J_u = \frac{dN_{[d_u,+\infty)}}{dt}$$
$$= \frac{N_{[d_u,d_u+dd)}\big|_{t+dt} + N_{[d_u+dd,+\infty)}\big|_{t+dt} - N_{[d_u,+\infty)}\big|_t}{dt}$$

$$= \frac{N_{[d_u,d_u+dd)}\big|_{t+dt}}{dt}$$

$$= \lim_{\Delta d \to 0} \frac{N_{[d_u,d_u+dd)}\big|_{t+dt}}{N_{[d_u+dd,d_u+\Delta d+dd)}\big|_{t+dt} \cdot dt} \cdot N_{[d_u+dd,d_u+\Delta d+dd)}\big|_{t+dt}$$
$$= \lim_{\Delta d \to 0} \frac{dd}{\Delta d \cdot dt} \cdot N_{[d_u,d_u+\Delta d)}\big|_t$$

$$= n_u \cdot GR_u \tag{A11}$$

Finally combining Eq. (A7) and Eq. (A11) we can obtain the equation to estimate nucleation rate as Eq. (A12),
$$I = \frac{dN_{[d_k,d_u)}}{dt} + \sum_{d_g=d_k}^{d_{u-1}} \sum_{d_i=d_{min}}^{+\infty} \beta_{(i,g)} N_{[d_i,d_{i+1})} N_{[d_g,d_{g+1})} - \frac{1}{2} \sum_{d_g=d_k}^{d_{u-1}} \sum_{\substack{d_i^3+d_{j+1}^3=d_g^3 \\ d_{i+1}^3+d_j^3=d_g^3 \\ d_i,d_j \geq d_{min}}} \beta_{(i,j)} N_{[d_i,d_{i+1})} N_{[d_j,d_{j+1})} + n_u \cdot GR_u . \tag{A12}$$

The first term in the RHS of Eq. (A12) is the change in the number concentration of particles ranged from $d_k$ to $d_u$. The second
and third terms are particle loss to coagulation scavenging and particle formation by coagulation, named as coagulation sink
term (*CoagSnk*) and coagulation source term (*CoagSrc*), respectively (Kuang et al, 2012). The fourth term is the condensational
growth term, which is an approximation of the formation rate, $J_u$. This balance formula derived from aerosol general dynamic
equation can also be expressed as Eq. (A13).
$$I = \frac{dN_{[d_k,d_u)}}{dt} + CoagSnk - CoagSrc + n_u \cdot GR_u \tag{A13}$$

When applying Eq. (A12) in practice, $d_k$ is usually the assumed size of the critical nuclei (or the lowest size limit of instrument,
corresponding to formation rate, $J_k$, rather than nucleation rate, $I$). The $dN/dt$ term can be obtained either by differentiating
between adjacent time bins or fitting in a continuous time period. *CoagSnk* and *CoagSrc* can be directly calculated from particle
size distribution, where $d_{min}$ is the minimum detected particle diameter. If formation by coagulation of smaller clusters is also
included in the definition of nucleation rate, calculation of *CoagSrc* (the third term in the RHS of equation A(12)) should begin
with $d_{k+1}$ instead of $d_k$, which usually affects little since the difference is only a size bin and the whole *CoagSrc* is usually a





minor term of $J$ in atmosphere environment. Growth rate can be estimated by different methods (Weber et al., 1996; Weber et
al., 1997; Kulmala et al., 2012; Lehtipalo et al., 2014), or the growth term can be simply neglect if $d_u$ is proper large.
It should be clarified that the formation rate calculated using Eq. (A12) may be underestimated because coagulation scavenging
by particles and clusters smaller than $d_{min}$ is neglected due to the limitation of measuring instruments. As illustrated in Fig.
6(a), *CoagSnk* calculated using $d_p$ larger than 3 nm is ~ 89.1% of that using $d_p$ larger than 1.5 nm. It could be inferred that the
calculated $J_3$ was slightly underestimated in some previous studies lacking size distribution for sub-3 nm particles. While in
this study, measured particles down to 1.3 nm are accounted for when calculating $J_{1.5}$ and $J_3$. Neglecting coagulation between
clusters may also have a non-negligible effect on the calculated results (McMurry 1983), which calls for measurement of major
molecular clusters participating in nucleation if more accurate formation rate is to be obtained.
**Appendix B**
**Relationships with previous approaches**
Several approaches have been previously proposed to estimate formation rate. Two widely used approaches are a definition
approach and a balance approach. Since the new balance approach proposed in this study is based on aerosol general dynamic
equation with a reasonable assumption that net coagulation of any particle larger than the "proper large" upper bound, $d_u$, and
other particles can be neglected, its inner relationships with former approaches can be elucidated by making additional
assumptions and approximations.
Formation rate is defined as the flux that particles grow pass through the given size, and can be expressed as Eq. (B1) (Heisler
& Friedlander, 1977; Weber et al., 1996; Kuang et al., 2008; Kuang et al., 2012). Note that Eq. (B1) is valid only when it is in
the continuous space of particle diameter, while a more accurate expression in the discrete form is shown as Eq. (B2).
$J_k = n_k \cdot GR_k$                                                                 (B1)
$J_k = n_{k-1} \cdot GR_{k-1}$                                                         (B2)
Eq. (B2) is believed to be theoretically correct since the only condensational flux into $d_k$ is the growth of smaller clusters or
particles with diameter of $d_{k-1}$. Although in similar expression with Eq. (A11), Eq. (B2) focuses on the flux into rather than out
of the size bin for calculation, and there's no need to account for coagulation scavenging, as illustrated in Fig. 1.
A theoretical expression of $GR$ proposed in previous study is shown as Eq. (B3), where $\alpha$ is herein the coagulation efficiency
(fraction of collisions that successfully result in coagulation), $V_1$ is the volume increment when adding a single gaseous
precursor, and $v$ is the mean thermal velocity of the gaseous precursor (Weber et al., 1996). Here we update the equation by
considering different chemical species and describing coagulation by $\beta$, as shown in Eq. (B4). The subscript c denotes different
chemical species of monomers participating in the condensational growth of cluster k-1, and $N_{1c}$ is their corresponding number
concentration. Coagulation efficiency is included in each $\beta_{(1c,k)}$ (Eq. 13.56, Seinfeld & Pandis 2006).



$$GR_k = \frac{\alpha V_1 N_1 v}{2} \qquad (B3)$$
$$GR_{k-1} = \frac{\sum_c \beta_{(1c,k-1)} N_{1c} N_{k-1}}{n_{k-1}} \qquad (B4)$$
Eq. (B2) is a mathematical truth, however, it faces difficulties when applying in practice, since $n_{k-1}$ is obtained by
approximation over some size range around $d_k$ rather than the true frequency density at cluster k-1, $dN_{k-1}/dd_{k-1}$. Moreover,
because size distribution smaller than $d_k$ is difficult to obtain, the size range for estimation is usually larger than $d_k$. For example,
the formula to estimate $J_3$ using nano-SMPS data in Kuang et al. (2008) is shown as Eq. (B5). Although Eq. (B5) seems to be
an estimation of Eq. (B2), they are essentially two different equations. This is because the measured particle number
concentration in the size range for calculation, i.e., $N_{3-6}$ in Eq. (B5), has been affected by coagulation. By comparing with Eq.
(A13), it can be concluded that $dN/dt$, *CoagSnk* and *CoagSrc* are simply neglected in Eq. (B5), while Eq.(B2) does not suffer
from this problem by its definition.
$$J_3 \approx \frac{N_{3-6}}{3\,\text{nm}} \cdot GR_{1-3} \qquad (B5)$$
There are also problems in estimating $GR_{k-1}$. Equation (B4) is only a theoretical formula, since it is nearly impossible to
determine all the chemical species contributing to nucleation and their corresponding coagulation coefficients in the
complicated atmospheric environment. *GR* calculated by sulfuric acid itself using Eq. (B3) may lead to underestimation (Kuang
et al., 2010), while uncertainties also exist in the approaches which fit particles size distribution to obtain *GR* (Kulmala et al.,
2012; Lehtipalo et al., 2014) because the effect of coagulation on measured size distribution is also neglected. So conclusively,
Eq. (B2) is considered to be theoretically correct, however, it's not recommend to be applied for analyzing NPF events with
high coagulation scavenging.
The other approach is a balance method based on a macroscopic point of view shown as Eq. (B6) (Kulmala et al., 2001;
Kulmala et al., 2004), and here we adopt the equation in the most recent paper (Kulmala et al, 2012). *CoagS* is named as
coagulation sink and defined by Eq. (B7), where the subscript m corresponds to the representative diameter, $d_m$, in the size
range from $d_k$ to $d_u$. Usually $d_m$ is the the geometric mean diameter of $d_k$ and $d_u$. However, coagulation of any particle smaller
than $d_m$ or even $d_u$ with other particles is sometimes neglected when it comes to calculation, such as the formula suggested in
Kulmala et al (2012) shown as Eq. (B8).
$$J_k = \frac{dN_{[d_k,d_u)}}{dt} + CoagS_m \cdot N_{[d_k,d_u)} + \frac{N_{[d_k,d_u)}}{(d_u-d_k)} \cdot GR_{[d_k,d_u)} \qquad (B6)$$
$$CoagS_m = \sum_{d_i=d_{min}}^{+\infty} \beta_{(i,m)} N_i \qquad (B7)$$
$$CoagS'_m = \sum_{d_i=d_m}^{+\infty} \beta_{(i,m)} N_i \qquad (B8)$$





Eq. (B6) appears similar to Eq. (A13) since they both originate from the population balance method, however, there are some
differences between them.
Firstly, the upper bound of particle size in Eq. (B6), $d_u$, is lack of strict definition and discussion. In relatively early literatures,
$d_u$ usually refers to the upper bound of nucleation mode particles, i.e., 25 nm (Kulmala et al., 2001; Dal Maso et al., 2005), or
theoretically defined as the maximum size the critical clusters can reach during a short time interval (Kulmala et al., 2004).
While in recent studies, the size range for estimation is usually reduced, e.g., to a upper bound of 6 nm (Sihto et al., 2006;
Riipinen et al., 2007; Passonen et al., 2009; Vuollekoshi et al., 2012). However, as discussed in Appendix A, $d_u$ should be
decided by the two criterions that effects of transport and primary emission are negligible and the condensational growth term,
$J_u$, is relative small compared to $J_k$. The upper bound of 25 nm is usually reasonable since high concentration of particle formed
by nucleation predominates the coagulation sink term during strong new particle formation time, while the upper bound of 6
nm may lead to underestimation when freshly formed particles grow beyond, as discussed in the main text.
Secondly, scavenging by coagulation with particles smaller than $d_m$ is not included if using Eq. (B8) to calculate $CoagS$. As
shown in Fig. B1, $CoagS$ is always larger than $CoagS'$, and their difference increases as $d_m$ increases. $CoagS'_{8nm}$ is ~31% of
$CoagS_{8nm}$, indicating a large amount of underestimation when using Eq. (B8). Note that Eq. (B7) and the approximation
formula (estimated with condensation sink) proposed by Lehtinen et al. (2007) does not suffer from this problem.
Thirdly, the second term in the RHS of Eq. (B6) is not always a reasonable approximation of $CoagSnk$ in Eq. (A12) and Eq.
(A13). Theoretically, the relationship between $CoagSnk$ and $CoagS$ is shown as Eq. (B9), while $CoagS_m$ is chosen as the
representative value when estimating $J$ using Eq. (B6).
$$CoagSnk = \sum_{d_g = d_k}^{d_u} CoagS_g \cdot N_g \qquad (B9)$$

However, neither is $CoagS$ a relatively constant value versus particle diameter nor is $CoagS_m$ the mean value of $CoagS$ in
calculated size range from $d_k$ to $d_u$. As illustrated in Fig. B1, coagulation coefficient with 8 nm particles decreases rapidly with
the increase in $d_i$ when particle is smaller than 8 nm. The minimum value of $\beta_{(d_i, 8nm)}$ appears at $d_i$ around 8 nm because
particles with similar thermal velocities are more difficult to collide with each other. The calculated $CoagS'$ during a strong
NPF event on Mar. 27th, 2016 appears monotonously decreasing with the increase of $d_m$, while the calculated $CoagS$ has a
minimum value at 6.7 nm because $CoagS$ is mainly attributed to nucleation mode particles during NPF events. In this example,
$CoagS_{8nm}$ and $CoagS'_{8nm}$ are ~22.6% and ~7.2% of $CoagS_{1.5nm}$, respectively, indicating non-negligible underestimation of
coagulation sink term as well as nucleation rate when using a constant $CoagS_m$ instead of a varying value (as a function of
particle diameter).



Fourthly, particle formation by coagulation is neglected in Eq. (B6). The absence of *CoagSrc* will lead to an overestimation of
nucleation rate. However, it sometimes coincidently cancels out with the underestimation caused by using *CoagS_m* to
approximate *CoagSrc*, as discussed in the main text.
Fifthly, the growth term in Eq. (B6) is estimated over the whole size range from $d_k$ to $d_u$, while in Eq. (A12) it is mathematically
restricted at the upper bound, $d_u$. $n_u$ is usually smaller than mean value in the size range from $d_k$ to $d_u$ during a NPF event, and
recent work have revealed that the observed *GR* is size dependent (Kuang et al., 2012; Kulmala et al., 2013; Xiao et al., 2015).
For example, as shown in Fig. B2, *GR* varies with time in the NPF event on Apr. 3rd, 2016, and was linearly fitted in different
diameter ranges. The mean *GR* of particles ranged from 2 nm to 25 nm is ~7.47 nm/h, while $GR_{25}$ is ~10.86 nm/h. At 11:30
on Apr. 3rd, $n_{25}$ (d$N$/dlog$d_p$ at 25 nm) is 164 #/cm$^3$, while the mean $n$ of particles ranged from 2 nm to 25 nm is 4755 #/cm$^3$.
The calculated condensational growth term in Eq. (B6) is ~20 times of that in Eq. (A12).
In relatively clean environment with weak NPF events, Eq. (B6) may work well since the calculated $J_k$ is mainly predominated
by d$N$/d$t$. However, when number concentration of aerosol formed by nucleation and (or) background aerosol is high, i.e.,
*CoagSnk* is the major component of $J_k$, Eq. (B6) underestimates the formation rate (and nucleation rate) due to underestimation
of the coagulation scavenging effect.

**Acknowledgement**
Financial supports from the National Science Foundation of China (21422703, 41227805 & 21521064) and the National Key
R&D Program of China (2014BAC22B00 & 2016YFC0200102) are appreciated.

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



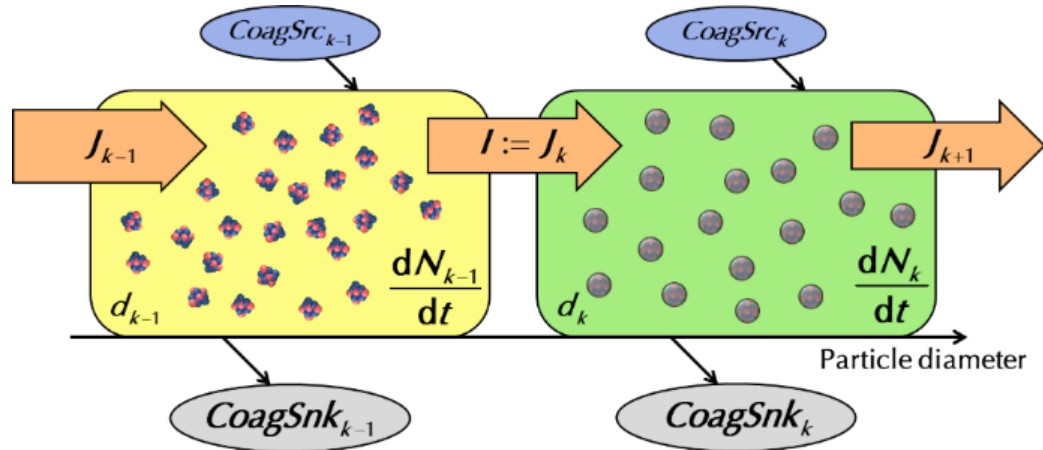


**Figure 1: Schematic of the general dynamic equation.**





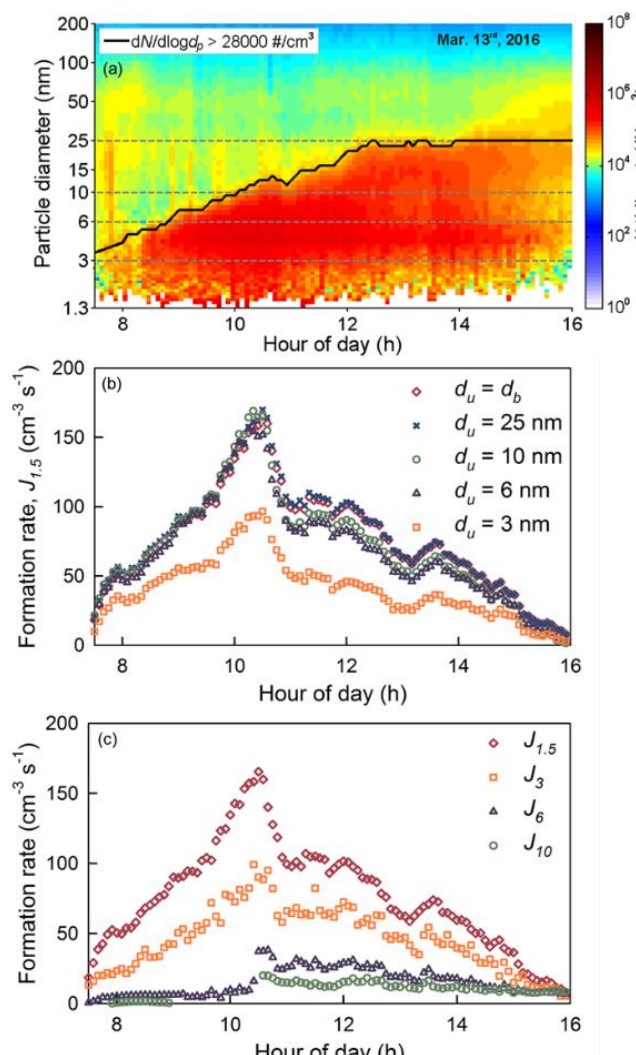

**Figure 2: Comparison of formation rates estimated using different upper bounds, $d_u$. (a) A typical new particle formation event. Dashed gray lines represent different $d_u$ in Eq. (1). Solid black lines corresponds to $d_b$, i.e., the varying upper bound determined by dN/dlog$d_p$. (b) Estimated formation rates with different upper bound, $d_u$, using Eq. (1). (c) Estimated formation rates with different $d_k$ using Eq. (1). $d_u$ equals to 25 nm and $d_{min}$ equals to 1.3 nm in the four scatter plots.**





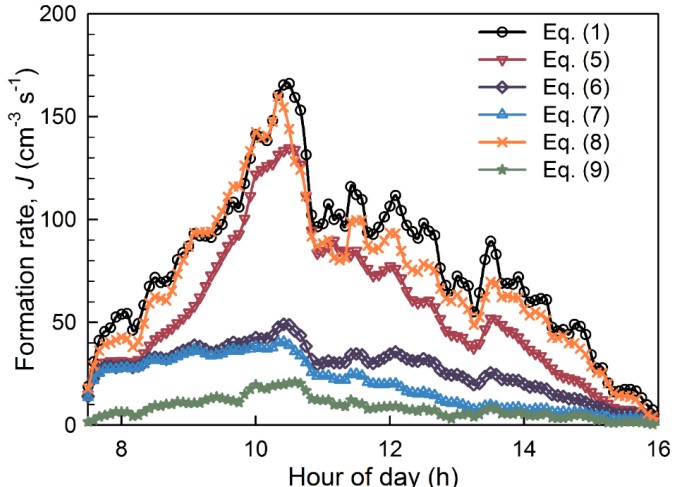

**Figure 3: Comparison of formation rates estimated by different formulae.**



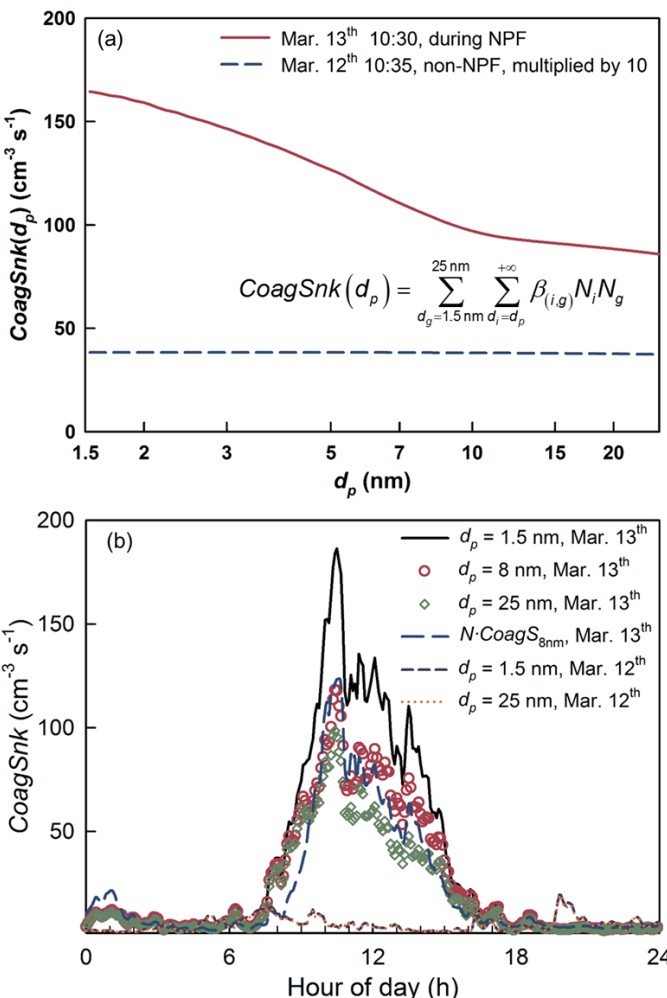


**Figure 4: (a)** *CoagSnk* **as a function of** $d_p$**, where** $d_p$ **is the accounted minimum diameter when calculating** *CoagS_g* **for particles at all**


**different** $d_g$**, and scavenging due to coagulation with particles small than** $d_p$ **is neglected, as the defined by the formula in panel (a).**


**The dashed line corresponding to** *CoagSnk* **on a non-NPF day is also monotonously decreasing with the increase of** $d_{min}$ **by a negligible**


**slope. (b) Time evolution of** *CoagSnk* **versus time on a NPF day (Mar. 13th) and a non-NPF day (Mar. 12th).** $d_p$ **is defined the same**


**with that in panel (a).** *N* **is the number concentration of particles in the size range from 1.5 nm to 25 nm, while** *CoagS_{8nm}* **is calculated**


**using Eq. (3).**







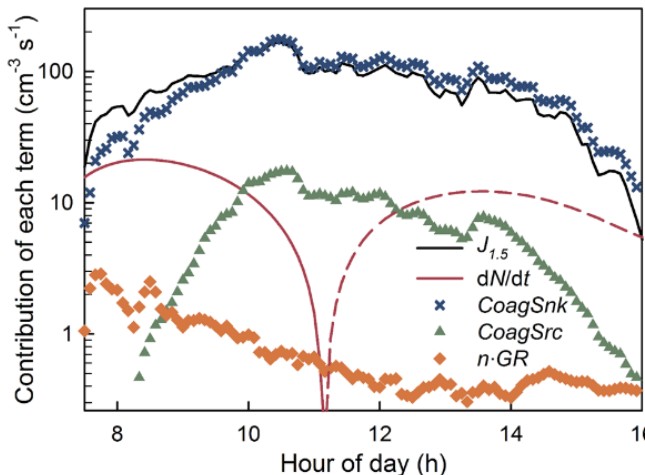


**Figure 5: Contribution of each term to the estimated formation rate. d$N$/d$t$ is obtained by fitting and shown in absolute value with**

**solid and dashed lines corresponding to positive and negative parts, respectively. Note the upper bound, $d_u$, equals $d_b$ as defined**

**section 4.1 for better accuracy, however, it doesn't affect the generality of the result.**






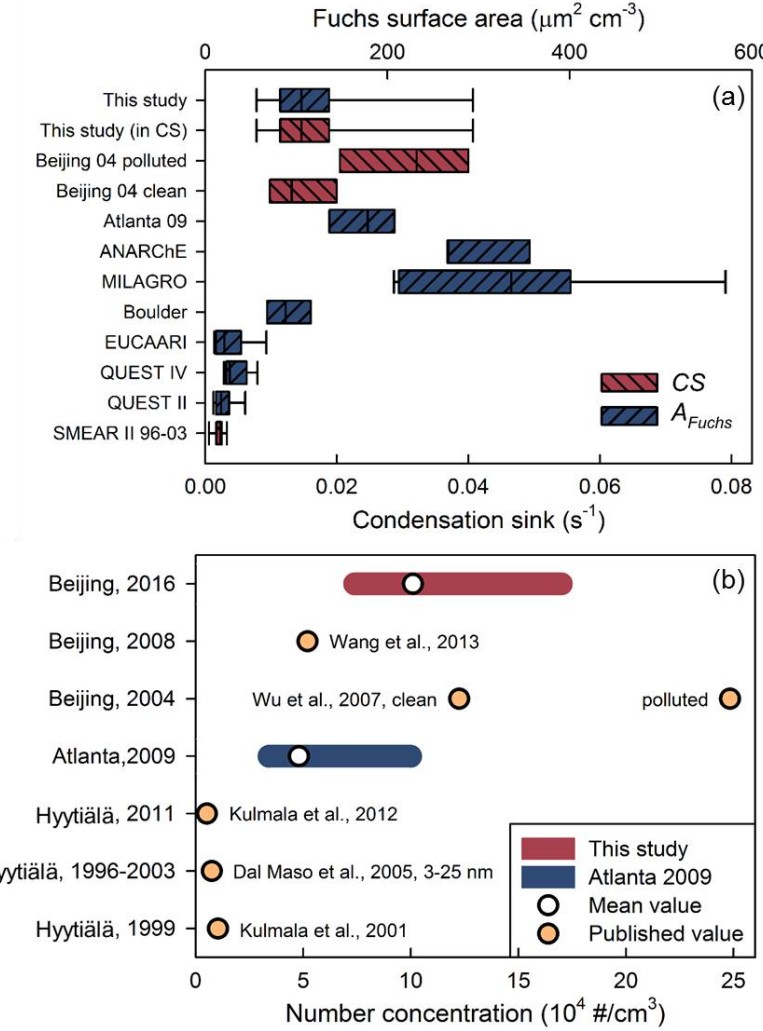

**Figure 6: (a) Comparison of Fuchs surface area and condensation sink in Beijing (when NPF events occurred) with those in other locations. NPF days was classified by condensation sink in urban Beijing in 2004 (Wu et al., 2007). ANARChE (Mcmurry et al., 2005) and MILAGRO (Iida et al., 2008) were conducted in Atlanta and Tecamac, respectively, while EUCCARI (Manninen et al., 2009), QUEST II (Sihto et al., 2006), QUEST IV (Riipinen et al., 2007) was conducted in SMEAR II (Dal Maso et al., 2005), Hyytiälä. $A_{Fuchs}$ data in MILAGRO, ANARChE, Boulder, EUCCARI, QUEST II, and QUEST IV were published in Kuang et al. (2010). The ends of coloured rectangular correspond quartiles, while error bar represents the 10th and 90th percent value. (b) Comparison of peak number concentration of particles larger than 3 nm during NPF events in this study with those in Atlanta and other published data. Note that the published values (light orange points) in previous studies are not necessarily the mean values of the whole campaign periods.**





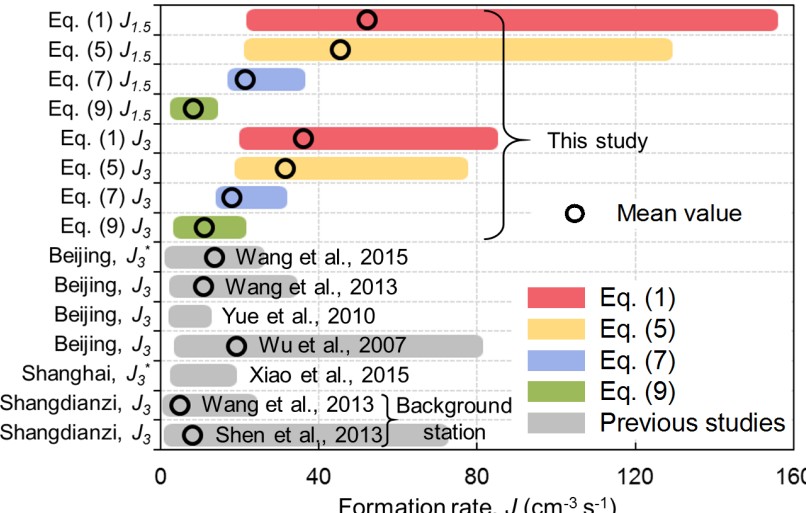


**Figure 7: Estimated $J_{1.5}$ and $J_3$ using different equations. Previously reported $J_3$ in China were included for comparison. The ends**

**of coloured rectangular correspond to the minimum value and the maximum value, respectively. *: The upper size bound to estimate**

**formation rate, $d_u$, is 6 nm (rather than 25 nm) in Wang et al., 2015 and Xiao et al., 2015.**






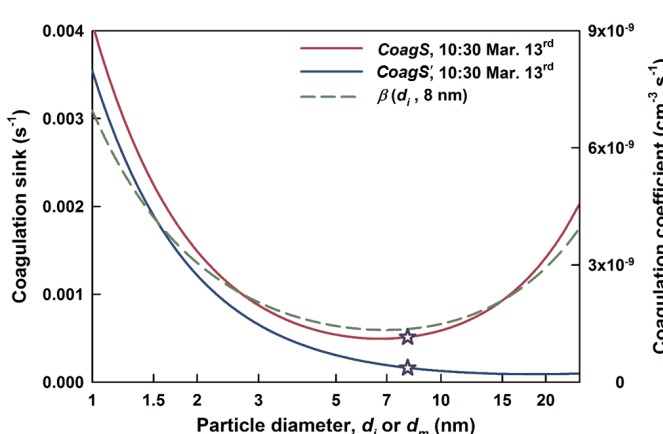


**Figure B1: Coagulation coefficient and calculated coagulation sink during a typical NPF event.** $CoagS$ **and** $CoagS'$ **are defined by**
**Eq. (B7) and Eq. (B8), respectively, and** $d_m$ **in this figure is treated as a variable rather than a constant value. The upper and lower**
**star denote** $CoagS'_{8nm}$ **and** $CoagS_{8nm}$ **which are used in the second terms in the RHS of Eq. (5) and Eq. (6) to approximate** $CoagSnk$,
**respectively.**



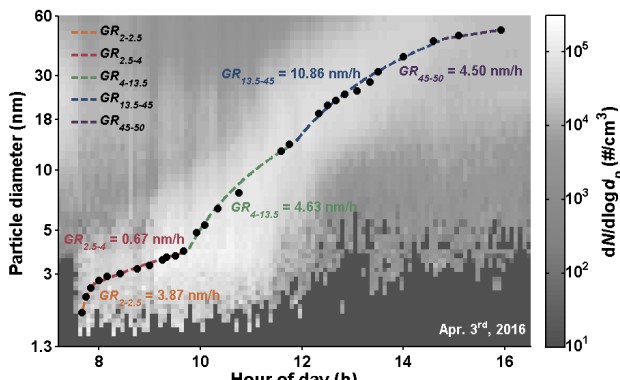


**Figure B2: Size and time dependent growth rate on a NPF day observed in Beijing. Representative diameters are obtained by**
**lognormal fitting of nucleation mode particles in each time bin, and *GR* is linearly fitted in each section.**