# Peer review of "A new balance formula to estimate new particle formation rate: reevaluating the effect of coagulation scavenging"

_Atmospheric Chemistry and Physics, 2017_

## Referee Comment (RC1) · Anonymous Referee #1 · 2 May 2017

**Summary:**

This work proposed a new formula to estimate particle formation rates during nucleation event. This method emphasizes the importance of coagulation scavenging effect among newly formed particles, especially in polluted urban area. The manuscript fits well to the scope of ACP and presents valuable methods/results. Thus I recommend it to be published after the following moderate/minor comments listed below have been adequately addressed.

**Comments:**

1.  In the abstract (page 1, lines 28-29): The authors stated or cited previous study (Guo et al., 2014): "Continuous growth of nucleated particles also provides increasing aerosol surface area for heterogeneous physicochemical processes, which may contribute to haze formation." Is this conclusion/statement correct? Although many NPF events were regularly followed by an increasing of particle mass, this only indicates a high abundance of condensable vapors in the atmosphere. Actually, the newly formed particles can only grow up to 60~100 nm. The contribution by particles smaller than 100 nm to total aerosol surface area is not significant. Maybe the large number of grown particles can, through the coagulation process, grow into a larger size range, where they could contribute more efficiently to the particle mass concentration or particle extinction. But I think it is not proper, or at least it may result in certain misunderstanding, to directly connect the NPF (or continuous growth of nucleated particles) with haze formation. Please consider this issue.

2.  In section 3 Experiment, did you consider the diffusion loss in the DMA and neutralizer, as recommended by Wiedensohler et al., (2012)?

3.  Lines 138-141, why you chose 28000 cm$^{-3}$ as a boundary, from visual expression or mathematic calculation? Could you explain it more? And in line 326, do you mean Fig. 2b?

4.  Since you measured particle number size distributions down to ~1.5 nm, could you compare the $J_{1.5}$ between direct measurements (e.g., results in Fig.2b) and estimated using the experimental formula (Eq.11 in Kulmala et al., 2012). It would be interesting to check the uncertainty using the experimental formula to estimate $J_{1.5}$ in strong nucleation events.

5. Maybe it is better to move all the equations in section 4.2 to section 2, and only discuss the comparison results here.

6. I think it is unfair to include Eq. 8 in the discussion of section 4.2. Based on what I understand, it uses the different upper bound (6 nm), which is much lower than the bound (25 nm) in other equations. The better agreement of estimated $J_{1.5}$ is due to the "closer" $CoagS_m$ ($d_m$ =4 nm) used in Eq.8. Could you recalculate $J_{1.5}$ using Eq. 8 with the same upper bound (25 nm) and different $CoagS_m$ ($d_m$ = 2,3,4… nm), how the result correlates with that of Eq. 1.

7. Please also include the New Delhi case (Kulmala et al., 2005) in Fig. 6a. It also showed high $CS$ values.

8. Did you observed any relatively clean nucleation events during this campaign? How about the comparison of different formula in clean case?

9. There are several grammar mistakes in the text, please carefully check.

**References**

Kulmala, M., Petäjä, T., Mönkkönen, P., Koponen, I. K., Dal Maso, M., Aalto, P. P., Lehtinen, K. E. J., and Kerminen, V. M.: On the growth of nucleation mode particles: source rates of condensable vapor in polluted and clean environments, Atmos Chem Phys, 5, 409-416, 10.5194/acp-5-409-2005, 2005.

Kulmala, M., Petäjä, T., Nieminen, T., Sipilä, M., Manninen, H. E., Lehtipalo, K., Dal Maso, M., Aalto, P. P., Junninen, H., Paasonen, P., Riipinen, I., Lehtinen, K. E. J., Laaksonen, A., and Kerminen, V.-M.: Measurement of the nucleation of atmospheric aerosol particles, Nat. Protoc., 7, 1651-1667, 10.1038/nprot.2012.091, 2012.

Wiedensohler, A., Birmili, W., Nowak, A., Sonntag, A., Weinhold, K., Merkel, M., Wehner, B., Tuch, T., Pfeifer, S., Fiebig, M., Fjäraa, A. M., Asmi, E., Sellegri, K., Depuy, R., Venzac, H., Villani, P., Laj, P., Aalto, P., Ogren, J. A., Swietlicki, E., Williams, P., Roldin, P., Quincey, P., Hüglin, C., Fierz-Schmidhauser, R., Gysel, M., Weingartner, E., Riccobono, F., Santos, S., Grüning, C., Faloon, K., Beddows, D., Harrison, R., Monahan, C., Jennings, S. G., O'Dowd, C. D., Marinoni, A., Horn, H. G., Keck, L., Jiang, J., Scheckman, J., McMurry, P. H., Deng, Z., Zhao, C. S., Moerman, M., Henzing, B., de Leeuw, G., Löschau, G., and Bastian, S.: Mobility particle size spectrometers: harmonization of technical standards and data structure to facilitate high quality long-term observations of atmospheric particle number size distributions, Atmos. Meas. Tech., 5, 657-685, 10.5194/amt-5-657-2012, 2012.

---

## Referee Comment (RC2) · Anonymous Referee #2 · 23 May 2017

Review of A new balance formula to estimate new particle formation rate: reevaluating the effect of coagulation scavenging

A new method to estimate particle formation rates has been proposed by the authors which is an improvement to existing ones with respect to taking coagulation into account. This seems to be an important improvement whan analysing new particle formation events in polluted conditions, such as Beijing, where the method is applied. Also, a nice comparison with some previous approaches is presented. The topic fits well to ACP and deserves publication, but some major modifications are first necessary. In addition, the manuscript suffers from several grammatical errors as well as unclear writing (some of the points I have commented below but many not). A thorough language-check is thus essential.

Comments: 1. Eq. 1 and Appendix A: I believe that eq. 1 is a direct consequence of the GDE and does not require the rather complicated derivation of Appendix A. If one starts with the continuous GDE, integrates it from $d\_k$ to $d\_u$, and finally writes the coagulation terms in discrete form with the bins - then equation 1 is self-evident (or can be derived in 3 lines)?

2. Eq. 1: The writing under the summation symbol is very small font and almost unreadable. It has to be made more clear. Also, does the validity of the equation require a bin structure such that $(d\_i)^3 + (d\_j+1)^3 = (d\_g)^3$ ?

3. Page 3, line 84: What does "not included" mean here?

4. Page 4, lines 112-113: Why is information about size distribution below $d\_k$ needed, when applying equation 4?

5. Page 4, discussion after eq. 4 and Page 7, lines 195-196: The main problem in eq. 9 is that GR and n are not estimated at 1.5 nm in an optimal way but above it, isn't it. If, instead of using the range 1.5 - 2.5, one would use 1 - 2, the result would be much better?

6. Page 5, lines 137-139: Why "varying upper size bound"? And why 28,000 cm-3? Furthermore, I don't understand the sentence "$d\_u$ is equal to 25 nm rather that $d\_b$ when calculating dN/dt..."???

7. How do you determine GR?

8. Appendix B, page 15, line 393: I am not sure that the statement "Equation B2 is a mathematical truth" is correct, especially if (k-1) refers to a (wide) bin? The physical process of condensation is by monomer additions.

9. Much of Appendix B is repetition of the main text. Can it be shortened and combined with the main text so that any Appendix would not be necessary?

10. It is really interesting to see the performance of the different approaches when applied to experimental data. The paper would, however, become even better if validation of the new method would be demonstrated with synthetic data, for which the answer is known. I am not saying that this is a necessity for this paper, but something for the authors to consider.

———————————————————

---

## Author Comment (AC1) · 15 Jun 2017

**Responses to Reviewers' Comments on Manuscript acp-2017-199**

**(A new balance formula to estimate new particle formation rate:**

**reevaluating the effect of coagulation scavenging)**

We thank the reviewers for their time, efforts, and thoughtful comments that help to improve this manuscript. We have addressed the comments in the following paragraphs and made corresponding changes in the revised manuscript. Comments are shown as blue italic text followed by our responses. Changes are highlighted in the revised manuscript and shown as bold text in the responses. Line numbers quoted in the following responses correspond to the revised manuscript. In addition, the format of references has been updated.

**Reviewer #1:**

*Comment:*

*This work proposed a new formula to estimate particle formation rates during nucleation event. This method emphasizes the importance of coagulation scavenging effect among newly formed particles, especially in polluted urban area. The manuscript fits well to the scope of ACP and presents valuable methods/results. Thus I recommend it to be published after the following moderate/minor comments listed below have been adequately addressed.*

*1) In the abstract (page 1, lines 28-29): The authors stated or cited previous study (Guo et al., 2014): "Continuous growth of nucleated particles also provides increasing aerosol surface area for heterogeneous physicochemical processes, which may contribute to haze formation." Is this conclusion/statement correct? Although many NPF events were regularly followed by an increasing of particle mass, this only indicates a high abundance of condensable vapors in the atmosphere. Actually, the newly formed particles can only grow up to 60~100 nm. The contribution by particles smaller than 100 nm to total aerosol surface area is not significant. Maybe the large number of grown particles can, through the coagulation process, grow into a larger size range, where they could contribute more efficiently to the particle mass concentration or particle extinction. But I think it is not proper, or at least it may result in certain misunderstanding, to directly connect the NPF (or continuous growth of nucleated particles) with haze formation. Please consider this issue.*

Response: We appreciate this suggestion and it was well taken. Statement and citation on haze formation were removed. The sentence was revised as "**Continuous growth of nucleated particles also provides increasing aerosol surface area for heterogeneous physicochemical processes.**"

*2) In section 3 Experiment, did you consider the diffusion loss in the DMA and neutralizer, as recommended by Wiedensohler et al., (2012)?*

Response: Yes, both diffusion losses in the DMA and neutralizer were considered during data inversion.

The sentence was revised as "**A C++ program was used to invert particle size distribution from raw counts while incorporating diffusion losses inside the sampling tube, diffusion losses and charging efficiencies of the bipolar neutralizers, penetration efficiencies and transfer functions of DMAs, and detection efficiencies of CPCs (Hagen and Alofs, 1983; Jiang et al., 2011a)**."

*3) Lines 138-141, why you chose 28000 cm$^{-3}$ as a boundary, from visual expression or mathematic calculation? Could you explain it more?*

Response: 28000 cm$^{-3}$ was determined visually, however, the estimated $J_{1.5}$ appears to be insensitive to the exact value of the boundary. We revised the relevant paragraph (pages 6-7, lines 163-173) to better illustrate this.

*4) And in line 326, do you mean Fig. 2b?*

Response: Thanks, corrected.

*5) Since you measured particle number size distributions down to ~1.5 nm, could you compare the $J_{1.5}$ between direct measurements (e.g., results in Fig.2b) and estimated using the experimental formula (Eq.11 in Kulmala et al., 2012). It would be interesting to check the uncertainty using the experimental formula to estimate $J_{1.5}$ in strong nucleation events.*

Response: We had done these comparisons before submitting this manuscript. The estimated $J_{1.5}$ using the semi-empirical formula proposed by Kerminen & Kulmala (2002) and updated in following studies (e.g., Eq.11 in Kulmala et al., 2012) were higher than those estimated using the new formula proposed in this study. We suspect that the fundamental assumptions made to derive the semi-empirical formula might be violated when coagulation sink is high. However, detailed derivations of the semi-empirical formula (including all the assumptions) are not readily available that make it difficult to reach conclusive findings. In addition, simulated NPF events will help to better address this issue.

*6) Maybe it is better to move all the equations in section 4.2 to section 2, and only discuss the comparison results here.*

Response: Thanks, we moved all the equations in section 4.2 to section 2.2.

*7) I think it is unfair to include Eq. 8 in the discussion of section 4.2. Based on what I understand, it uses the different upper bound (6 nm), which is much lower than the bound (25 nm) in other equations. The better agreement of estimated $J_{1.5}$ is due to the "closer" CoagS$_m$ ($d_m$ =4 nm) used in Eq. 8. Could you recalculate $J_{1.5}$ using Eq. 8 with the same upper bound (25 nm) and different CoagS$_m$ ($d_m$ = 2,3,4… nm), how the result correlates with that of Eq. 1.*

Response: "Closer" does not necessarily mean better (as shown in Fig. R1, $J_{1.5}$ was overestimated when $d_m = 3$ nm). Estimated $J_{1.5}$ using different $d_m$ with the same upper bound differ with each other in *CoagSnk* only, and the relationship between *CoagSnk* and *CoagS* has been illustrated in Eq. (B9).

$$CoagSnk = \sum_{d_g=d_k}^{d_u} CoagS_g \cdot N_g$$

(B9)

As shown in Fig. R1, the estimated $J_{1.5}$ using Eq. (5) when $d_m = 4$ nm was generally in good agreement with that estimated using Eq. (1). However, their relative ratio gradually increased with time. This is because that newly formed particles gradually grow into large particles and larger particles correspond to smaller *CoagS*. Since *CoagSnk* is affected by particle size distributions, the attempt to simplify the expression of *CoagSnk* using a constant *CoagSm* will result in uncertainties in the estimated $J_{1.5}$. Thus, we suggest not to use a single *CoagSm* to estimate *CoagSnk*, especially during intense NPF events.

[Figure]

**Figure R1: Comparison of formation rates estimated using Eq. (5) with different $d_m$. The estimated $J_{1.5}$ using Eq. (1) is plotted as a reference.**

We include Eq. (8) because it was widely used in previous studies. We revised the discussions on Eq. (8) as "**$J_{1.5}$ estimated using Eq. (8) agreed well with that estimated using Eq. (1), however, it does not mean that 6 nm serve as a better upper size bound than 25 nm. The good agreement between results estimated using Eq. (1) and (8) is because that the estimation of *CoagSnk* tends to be more accurate when using an average *CoagSm* in a narrower size range and the underestimation in this case is coincidently cancelled out by the overestimation of formation rate caused by neglecting *CoagSrc*.**"

(page 8, lines 208-211).

*8) Please also include the New Delhi case (Kulmala et al., 2005) in Fig. 6a. It also showed high CS values.*

Response: We updated Fig. 6a and related discussions to include New Delhi case.

*9) Did you observed any relatively clean nucleation events during this campaign? How about the comparison of different formula in clean case?*

Response: Although NPF events occurred on comparatively clean days, no clean (weak) NPF events were observed in Beijing during this campaign. We tested the new formula using NPF data observed in the clean atmosphere of Tibet. In this case, *CoagSnk* is a minor term in the estimated $J_2$ and thus Eq. (5)–(8) generally agreed well with Eq. (1). At the end of Appendix B, it is stated that as "In relatively clean environment with weak NPF events, Eq. (B6) may work well since the calculated $J_k$ is mainly predominated by d$N$/d$t$. (page 17, lines 458-459)"

*10) There are several grammar mistakes in the text, please carefully check.*

Response: Thanks, we carefully checked the manuscript again.

**Reviewer #2:**

*Comment*:

*A new method to estimate particle formation rates has been proposed by the authors which is an improvement to existing ones with respect to taking coagulation into account. This seems to be an important improvement when analysing new particle formation events in polluted conditions, such as Beijing, where the method is applied. Also, a nice comparison with some previous approaches is presented. The topic fits well to ACP and deserves publication, but some major modifications are first necessary. In addition, the manuscript suffers from several grammatical errors as well as unclear writing (some of the points I have commented below but many not). A thorough language-check is thus essential.*

*1) Eq. 1 and Appendix A: I believe that Eq. 1 is a direct consequence of the GDE and does not require the rather complicated derivation of Appendix A. If one starts with the continuous GDE, integrates it from $d_k$ to $d_u$, and finally writes the coagulation terms in discrete form with the bins - then Eq. 1 is self-evident (or can be derived in 3 lines)?*

Response: We disagree with the reviewer on this comment. The continuous GDE is shown in Eq. (R1)

(Friedlander, 2000; Kuang et al., 2012),

$$J_k = J_{k+1} + \frac{\mathrm{d}N_{[k,k+1]}}{\mathrm{d}t} + \int_{d_{\min}}^{d_{k+1}} n_j \mathrm{d}d_j \int_{d_k}^{d_{k+1}} \beta_{(i,j)} n_i \mathrm{d}d_i - \frac{1}{2} \int_{d_k}^{d_{k+1}} d_j^2 \mathrm{d}d_j \int_{d_{\min}}^{d_j} \beta_{(i,\bar{i})} n_i \overline{n_i} \frac{\mathrm{d}d_i}{\overline{d_i}^2} \quad , \quad (\text{R1})$$

$J_{k+1}$ still remains unsolved such that derivation presented in Eq. (A4)-(A5) is essential. Note that although Eq. (B1) is correct in the continuous space of particle diameter (as shown in Fig. R2), one can not simply substitute it into Eq. (R1), since Eq. (B1) underestimates $J_k$ in the discrete space as illustrated in Fig. 1 and Eq. (B2).

[Figure]

**Figure R2: Schematic of the general dynamic equation in the continuous form.**

If starting with a size region in a macroscopic point of view, one can only obtain approximate formulae such as Eq. (5) that was originally reported by Kulmala et al., 2001.

It should be clarified that GDE in the continuous form is only an approximation of that in the discrete form. The approximation tends to be accurate when $d_p \gg d_k$ (the size of the critical nuclei). That is the reason why we started with the discrete GDE and stated that "Practically, Eq. (A7) is only an estimation of Eq. (A5)" (page 12, line 323) when converting it into the continuous form.

We believe that Appendix A is one of the essential parts of this study. Only by knowing detailed derivations and assumptions made during the derivations, one can better understand and evaluate different approaches.

*2) Eq. 1: The writing under the summation symbol is very small font and almost unreadable. It has to be made more clear. Also, does the validity of the equation require a bin structure such that $(d_i)\hat{}3 + (d_{j+1})\hat{}3$ be= $(d_g)\hat{}3$ ?*

Response: We increased font size for Eq. (1) and (A7) to make them more readable. We also added the following sentence "**The defined relationships,** $d_i^3 + d_{j+1}^3 = d_g^3$ **and** $d_i^3 + d_{j+1}^3 = d_g^3$ **, are to assure that all particles smaller than $d_\mathbf{g}$ are accounted for only once.**" (page 12, lines 326-327).

*3) Page 3, line 84: What does "not included" mean here?*

Response: We revised the descriptions as "**particles with diameters of $d_u$ are not accounted for**" in page 3, line 83 and other places.

*4) Page 4, lines 112-113: Why is information about size distribution below d_k needed, when applying equation 4?*

Response: Because in the discrete GDE,

$$J_k = n_{k-1} \cdot GR_{k-1}.$$
(B2)

Please refer to page 15, lines 401-408 for detailed discussions.

We added "**(See Appendix B)**" in lines 113-114.

*5) Page 4, discussion after Eq. 4 and Page 7, lines 195-196: The main problem in Eq. 9 is that GR and n are not estimated at 1.5 nm in an optimal way but above it, isn't it. If, instead of using the range 1.5-2.5, one would use 1 - 2, the result would be much better?*

Response: Yes. The result using 1-2 nm was closer to but still lower than that estimated by Eq. (1). Note that the following issues still remain:

a) The coagulation effect is still dominating in the 1-2 nm size range. Net coagulation effect is not equal to zero when estimating $J_{1.5}$ using the size range of 1-2 nm. Simply neglecting *CoagSnk* will lead to an

underestimation.

b) The continuous GDE is out of function at $d_k$ if $d_k$ is the critical size, i.e., Eq. (1) and Eq. (9) using the size range of 1-2 nm both lead to uncertainties when estimating $J_k$. However, since $J_k$ is partially estimated using particles much larger than $d_k$ (at which the continuous GDE is almost accurate) in Eq. (1), its result is relatively credible.

c) The 50% cut-off size of the prototype DEG-UCPC used in this study was ~1.4 nm (mobility diameter) and the minimum detected atmospheric particle size was ~1.2 nm. Practically, uncertainties of the measured particle size distributions become higher for particles smaller than 1.5 nm.

Therefore, we did not discuss the result using the size range of 1-2 nm. Eq. (9) using the size range of 1.5-2.5 nm was compared and discussed because similar formula (e.g., Eq. (B5)) was used in previous studies.

*6) Page 5, lines 137-139: Why "varying upper size bound"? And why 28,000 cm⁻³? Furthermore, I don't understand the sentence "$d_u$ is equal to 25 nm rather that $d_b$ when calculating dN/dt..."???*

Response: We tried to minimize the influence of background aerosols by determining a upper size boundary for new particles. The maximum size of particle formed due to nucleation increases with time, so the upper size bound, $d_b$, varies with time.

28000 cm$^{-3}$ was determined visually. It is an approximate boundary as shown in Fig 2(a). The estimated $J_{1.5}$ was found to be insensitive to the exact value of the boundary.

When estimating $J_{1.5}$ using varying upper size bound, the d$N$/d$t$ term was calculated using fixed size range, since the influence on particle number concentration due to varying size range might be larger than that of background aerosols.

We revised section 4.1 (page 6, lines 163-173) to better illustrate these issues.

*7) How do you determine GR?*

Response: in page 5, lines 124-125, we added "**Growth rates in all formulae in section 2.2 were estimated using the mode-fitting method suggested in Kulmala et al. (2012).**"

*8) Appendix B, page 15, line 393: I am not sure that the statement "Equation B2 is a mathematical truth" is correct, especially if (k-1) refers to a (wide) bin? The physical process of condensation is by monomer additions.*

Response: k-1 refers to the specific diameter $d_{k-1}$, corresponding to particles containing k-1 molecules

(one molecule less than that of the critical nuclei). Size bins in this study are referred as half-open

intervals, e.g., $[d_i, d_{i+1})$ in Eq. (1).

We added "**where k is the number of molecules contained by the particle**" in page 15, lines 384-

385, and replaced "a mathematical truth" with "**theoretically correct**" in page 15, line 401.

*9) Much of Appendix B is repetition of the main text. Can it be shortened and combined with the main*
*text so that any Appendix would not be necessary?*

Response: We removed the repetition part in Appendix B. The purpose of Appendix B is to illustrate

differences between the new formula and previous approaches. Estimated formation rates using different

formulae are concisely compared in the main text, while appendix B helps to better illustrate these

comparisons. In addition, minor differences between Eq. (B6) and the new formula are also presented in

Appendix B. Thus, we prefer to keeping the shortened Appendix B.

*10) It is really interesting to see the performance of the different approaches when applied to*
*experimental data. The paper would, however, become even better if validation of the new method would*
*be demonstrated with synthetic data, for which the answer is known. I am not saying that this is a*
*necessity for this paper, but something for the authors to consider.*

Response: We appreciate this suggestion. Simulated data for both weak and intense NPF events help to

further evaluate different approaches. It can be a topic of following study.

---

## Author Response (AR2)

**Responses to Reviewers' Comments on Manuscript acp-2017-199**

**(A new balance formula to estimate new particle formation rate:**

**reevaluating the effect of coagulation scavenging)**

We thank the reviewer for checking the detailed derivation of the formula. We have addressed the comments in the following paragraphs and made corresponding changes in the revised manuscript.

*1.) The authors have done a good job in answering most of my comments and those of the other reviewer. I still, however, think that additional explanations are needed to a couple of my initial concerns.*
*1. Derivation of A7 and A6 in the Appendix and double summation term in eq. A7:*
*Isn't the message in A7 self-evident? I.e. the rate of change of particle concentration in range k....u is equal to condensational growth into range (I) minus condensation out of range ($J_u$) plus coagulation into range minus coagulation out of range ?*

Response: We agree with the reviewer that Eq. A7 is conceptually self-evident providing that aerosol general dynamic equation (GDE) can be expressed in the continuous form. However, previous studies presented this relationship differently (e.g., Eq. 5-8) such that they underestimate new particle formation rates when analyzing intense NPF events in the polluted atmosphere.

*2.) In A6, the rate of change of particle concentration above size k is equal to condensational growth into the range (I) minus collision rate in the range?*

Response: Yes. Here k is the critical cluster size (line 303). Equation A6 is mathematically simplified from Eq. A5 given that u is large enough. The rate of change of particle concentration for the whole aerosol population is equals to the formation rate of new particles minus the coagulation loss rate.

*3.) In my view, the only slightly difficult part to formulate is the coagulation into range term in A7, and this is not thoroughly explained here (and perhaps, slightly wrong?). Let's say that we have a linear bin structure in volume so that $v_1 = 1$, $v_2 = 2$, ....., $v_k = k$, ..... and we are looking at the range from 7 to 10, i.e. $dN(7,10)/dt$. According to the indexing in eq. A5, the following index-pairs contribute to the coagulation source-term into the range: 1 and 5, 2 and 4, 3 and 3, 4 and 2, 5 and 1, 1 and 6, 2 and 5, 3 and 4, 4 and 3, 5 and 2, 4 and 3, 1 and 7, 2 and 6, 3 and 5, 4 and 4, 5 and 3, 6 and 2, 7 and 1. The factor (1/2) takes care of the double counting and this is correct. But what about the 5 first pairs in the list? They also produce particles into the range 6 to 7 which is not in the range 7 to 10 ? Please explain! Also, what if you have a different bin structure, say logarithmically spaced? Then, the conditions under the summation term on the third term of the right hand side of equation A7, i.e. $v(i) + v(j+1) = v(g)$ cannot hold? If the equation in its current form is only applicable for a linearly spaced bin structure, it must be*

Response: The coagulation source term in Eq. 7 overestimates the formation rate due to coagulation. Since the continuous form GDE is expressed in the summation form (rather than the integral form), however, *CoagSnk* and *CoagSrc* have to be either overestimated or underestimated. To clarify this, we added "**The third term in the RHS of Eq. A7, i.e., the coagulation source term, is slightly overestimated. Since the coagulation source term is usually much less than the coagulation sink term when $d_u$ is limited (e.g., $d_u$ <50 nm), however, this overestimation can be neglected.**" (lines 327-329). When $d_u$ is infinite, the estimated *CoagSrc* is almost exactly the half of the estimated *CoagSnk* using Eq. A7.

*4.) In my original review, I suggested removing most of the derivations in the Appendix. As they are located in the Appendix, they might as well stay there, if the authors wish so.*

Response: We prefer to keeping the derivations in the Appendix. The above comments raised by the reviewer perfectly illustrates the necessity to include the derivations and key assumptions made.

---

## Author Response (AR3)

**Responses to Reviewers' Comments on Manuscript acp-2017-199**

**(A new balance formula to estimate new particle formation rate)**

We appreciate the comments and suggestions from the editor. They were addressed below and revisions of the manuscript were made accordingly. Equation numbers have been updated and they are quoted in the following responses correspond to the revised manuscript.

*After reading the revised paper and final responses to referee comments, I still have a few minor issues to be considered:*

*1) Response to the third comment by the referee: I do not think it is enough to say that the third term is overestimated. The authors should, at the very least, explain shortly why this term is overestimated. Also, It would be very good to know what "slightly" mean in this context. Are we taking about differences <1%, a few %, <10 % or what?*

Response: Eq. (A5) was written in the discrete form. When applying it in measured particle size distributions, it needs to be converted into the continuous form. Approximations are needed during the conversion. We agree with the referee that the conversion used in the previous version of our manuscript is affected by the bin structure and the estimated coagulation source has some deviation. In the last response, we argued that its contribution to the estimated $J_{1.5}$ is negligible since the coagulation source only accounts for a minor proportion to the estimated formation rate. For instance, the resultant uncertainty in the estimated daily maximum $J_{1.5}$ was less than 3.7%.

To address this question, we did further analysis and made additional revisions to the manuscript. A new conversion from the discrete form to the continuous form was derived and reported in the revised manuscript. This new conversion is not affected by the bin structure. Though this new conversion does not change the main findings in this manuscript due to the reason mentioned above, it serves as a method to properly estimate the coagulation effect from measured particle size distributions.

The revisions are given below (lines 324-332 in the revised manuscript):

**"For the third term in the RHS of Eq. (A5), i.e., the coagulation source term, its summation sequence can be rearranged as:**

$$
\begin{aligned}
\frac{1}{2}\sum_{g=k}^{u-1} \sum_{\substack{i+j=g \\ i,j\geq 2}} \beta_{(i,j)} N_i N_j \\
= \frac{1}{2}\beta_{(2,k-2)} N_2 N_{k-2} + ... + \frac{1}{2}\beta_{(k-2,2)} N_{k-2} N_2 \\
+ ... \\
+ \frac{1}{2}\beta_{(2,u-3)} N_2 N_{u-3} + ... + \frac{1}{2}\beta_{(k-2,u-k-1)} N_{k-2} N_{u-k-1} + ... + \frac{1}{2}\beta_{(u-3,2)} N_{u-3} N_2 \\
= \frac{1}{2}\sum_{g=2}^{u-3} \sum_{i=\max(2,k-g)}^{i+g \leq u-1} \beta_{(i,g)} N_i N_g
\end{aligned}
\tag{A7}
$$

The formulae in both the far LHS and the far RHS of Eq. (A7) are equally accurate to estimate the coagulation source term. However, simply substituting the continuous particle diameter (e.g., $d_g$) for the discrete size (e.g., $g$) in the far LHS of Eq. (A7) will result in uncertainties when the size bins do not increase linearly in the particle volume space. As indicated in Fig. A1, substituting the continuous particle diameter for the discrete size in the far RHS of Eq. (A7) is independent of the bin structure for $d_g$ and $d_i$.

Thus, Eq. (A5) can be rewritten as,

$$I = \frac{\mathrm{d}N_{[d_k,d_u]}}{\mathrm{d}t} + \sum_{d_g=d_k}^{d_{u-1}} \sum_{d_i=d_{\min}}^{+\infty} \beta_{(i,g)} N_{[d_i,d_{i+1}]} N_{[d_g,d_{g+1}]} - \frac{1}{2} \sum_{d_g=d_{\min}}^{d_{u-1}} \sum_{d_i^3=\max(d_{\min}^3,d_k^3-d_{\min}^3)}^{d_{i+1}^3+d_{g+1}^3 \le d_u^3} \beta_{(i,g)} N_{[d_i,d_{i+1}]} N_{[d_g,d_{g+1}]} + J_u$$

(A8).

[Figure]

Figure A1: Schematic for two different summation sequences to estimate the coagulation source term. Equations in panels (a) and (b) correspond to the continuous forms of the far LHS and the far RHS formulae in Eq. (A7), respectively. The coagulation source term is denoted by half the area of the triangle (since the particles at the same diameter are accounted for twice). The colored areas are the estimated area using the two equations, respectively.

*2) Lines 158-159: I am not sure you can call accommodation coefficient and coagulation efficiency the same because the former refers to condensation and the latter to coagulation.*

Response: Thanks for the suggestion. We revised it as "**Mass accommodation coefficient**" in line 158.

*3) Line 245: Can you measure the nucleation intensity simply by looking at particle number concentration? What is primary particle number emissions are very large, as they can be in polluted cities? Would change rates in number concentrations be a better measure? Please open this a bit.*

Response: During typical NPF events, the sharp increase in total particle number concentration is mainly due to nucleation and the contribution of primary emissions is negligible. In cases that primary emissions

are significant, a high abundance of particles in various sizes is often identifiable in the 3-D contour plots of particle size distribution data. We agree that the change rate in particle number concentration, d$N$/d$t$, is good measure of nucleation intensity. The quantity of particles produced by nucleation can also serves as an indicator. The aim of this paragraph is to explain the reason for the dominant role of *CoagSnk* in urban Beijing. The expression of *CoagSnk* is determined by particle number concentration rather than its change rates. To better illustrate this, we revised it as "**As shown in Equation (1), CoagSnk is approximately proportional to the square of particle number concentration. Nucleation intensity in urban Beijing, characterized by number concentration of particles larger than 3 nm during typical NPF event periods, is found to be higher than those in Hyytiälä and Atlanta (as shown in Fig. 6(b)).**" (lines 246-268)

*4) Grammatical comments:*

*The article (the) is missing from several places where it should be (lines 9, 10, 14, 16, 32, 41, 44, 70, 124, 157, 167, 175, 223, 236, 238, 254, 255, 270, 273, 274*

*criterions -- > criteria (lines 9, 92, 193, 263)*

*Line 14: ... and formulae used widely in the literature.*

*line 37: has -- > had*

*line 40: ...before they grow into larger sizes.*

*line 53: concentrations*

*line 66: ...narrower size ranges, such as ...*

*line 147: ... there are no tall buildings nearby.*

*line 177: This indicates that the influence…*

*line 180: beyond what?*

*line 184: The reason for*

*line 192: what is meant by "proper large"? properly large?*

*line 209: bad wording ( is because that the)*

Response: We appreciated detailed editing from the editor that helps to improve this manuscript. We have made the suggested changes. In addition, we read the manuscript again to correct grammar errors.